# Ribosome profiling reveals pervasive and regulated stop codon readthrough in *Drosophila melanogaster*

**Joshua G Dunn[1,2,3,4], Catherine K Foo[1,2,3], Nicolette G Belletier[5], Elizabeth R Gavis[5], Jonathan S Weissman[1,2,3,4]\***

[1]California Institute of Quantitative Biosciences, San Francisco, United States; [2]Department of Cellular and Molecular Pharmacology, University of California, San Francisco, San Francisco, United States; [3]Howard Hughes Medical Institute, University of California, San Francisco, San Francisco, United States; [4]Center for RNA Systems Biology, Berkeley, United States; [5]Department of Molecular Biology, Princeton University, Princeton, United States

**Abstract** Ribosomes can read through stop codons in a regulated manner, elongating rather than terminating the nascent peptide. Stop codon readthrough is essential to diverse viruses, and phylogenetically predicted to occur in a few hundred genes in *Drosophila melanogaster*, but the importance of regulated readthrough in eukaryotes remains largely unexplored. Here, we present a ribosome profiling assay (deep sequencing of ribosome-protected mRNA fragments) for *Drosophila melanogaster*, and provide the first genome-wide experimental analysis of readthrough. Readthrough is far more pervasive than expected: the vast majority of readthrough events evolved within *D. melanogaster* and were not predicted phylogenetically. The resulting C-terminal protein extensions show evidence of selection, contain functional subcellular localization signals, and their readthrough is regulated, arguing for their importance. We further demonstrate that readthrough occurs in yeast and humans. Readthrough thus provides general mechanisms both to regulate gene expression and function, and to add plasticity to the proteome during evolution.

\*For correspondence: weissman@cmp.ucsf.edu

**Competing interests:** The authors declare that no competing interests exist.

**Reviewing editor**: Nahum Sonenberg, McGill University, Canada

## Introduction

Upon encountering a stop codon, ribosomes can terminate translation with remarkable fidelity, yet they do not always do so. Stop codon readthrough, the decoding of a stop codon as a sense codon by the ribosome, plays important regulatory roles. Most immediately, readthrough diversifies the proteome by creating a pool of C-terminally extended proteins. In this capacity, it is essential to a variety of plant and animal viruses (*Cimino et al., 2011*; *Li and Rice, 1989*; *Napthine et al., 2012*; *Skuzeski et al., 1991*; *Yoshinaka et al., 1985*; reviewed in *Beier and Grimm, 2001*; *Firth and Brierley, 2012*). In eukaryotic host genes, readthrough is functionally important insofar as it may suppress pathological pheno-types caused by premature stop codons (*Kopczynski et al., 1992*; *Fearon et al., 1994*), antagonize nonsense-mediated decay (*Keeling et al., 2004*), and, by changing the C-terminal sequence of a given protein, modulate its activity (*Torabi and Kruglyak, 2012*), stability (*Namy et al., 2002*), and/or localization (*Freitag et al., 2012*). In yeast, the efficiency of translation termination is modulated by [*PSI*⁺], an epigenetic state resulting from prion-like aggregates of Sup35p, the yeast homologue of the translation termination factor eRF3 (reviewed in *Tuite and Cox, 2007*). Various yeast strains exhibit [*PSI*⁺]-dependent fitness advantages, implying that increased readthrough activates useful genetic diversity that is ordinarily masked by stop codons (*True and Lindquist, 2000*; *Halfmann et al., 2012*). In addition, a small baseline level of readthrough appears to be beneficial in wild [*psi*⁻] yeast

**eLife digest** For a gene to give rise to a protein, its DNA is first used as a template to produce a messenger RNA molecule. Each group of three nucleotides within the messenger RNA encodes an amino acid, and structures called ribosomes assemble the protein by joining together amino acids in the correct order. The nucleotide triplets are called codons, and some are known as stop codons because they typically instruct the ribosome to stop adding amino acids.

Sometimes ribosomes interpret stop codons as amino acid insertion signals, giving rise to an extended protein with a modified structure or function. This phenomenon is known as stop codon readthrough, and is required for many viruses to complete their reproductive cycles. However, much less is known about stop codon readthrough in other organisms.

Now, Dunn et al. have used a technique called ribosome profiling to analyze stop codon readthrough across the entire genome of the fruit fly *Drosophila melanogaster*. An enzyme was used to fragment messenger RNA, and those fragments that were specifically engaged by ribosomes—and thus likely to encode protein—were sequenced. Stop codon readthrough occurred much more often than had been expected based on previous studies. Indeed, computational analysis strongly suggests that evolution has favored this process for certain fruit fly genes. Moreover, stop codon readthrough was also observed in yeast and human cells, suggesting that it is important in many organisms, not just the fruit fly.

Stop codon readthrough thus provides a novel way for organisms to tune the expression levels and functions of their genes, both throughout the lifetime of an individual, and the evolution of a species.

strains, as alleles of various factors controlling termination efficiency are under balancing selection (*Torabi and Kruglyak, 2011*).

However, a broad understanding of the biological roles of readthrough in eukaryotes remains elusive due to a lack of experimental data. To date, only a handful of eukaryotic host genes have been experimentally demonstrated to undergo readthrough in wild-type or prion-free organisms (*Geller and Rich, 1980*; *Xue and Cooley, 1993*; *Klagges et al., 1996*; *Steneberg et al., 1998*; *Namy et al., 2002*; *Jungreis et al., 2011*; *Freitag et al., 2012*; *Torabi and Kruglyak, 2012*; *Yamaguchi et al., 2012*). Compelling evidence that readthrough is broadly important in eukaryotes came with the development of algorithms (CSF and PhyloCSF) that use orthologous nucleotide sequences from related organisms to identify protein-coding regions of a reference genome based upon signatures of amino acid conservation (*Lin et al., 2007*, *2011*). Using this approach, 283 readthrough events were predicted in *Drosophila melanogaster*, six of which they confirmed experimentally (*Lin et al., 2007*; *Jungreis et al., 2011*). While these algorithms provide a powerful means to identify ancient and phylogenetically conserved readthrough events, they are limited in their ability to detect evolutionarily recent events. Nor can bioinformatic approaches identify a priori the tissues or cell types in which readthrough occurs, measure the fraction of ribosomes that read through a given stop codon, or determine whether any of these processes are regulated: such questions demand experimental approaches.

To this end, we present a modified ribosome profiling protocol—based on the deep sequencing of ribosome-protected footprint fragments (*Ingolia et al., 2009*)—that enables analysis of translation at a genome-wide level in *D. melanogaster*. Application of the *Drosophila* ribosome profiling strategy allows annotation of the *Drosophila* proteome using empirical data. By examining the physical locations of ribosomes along mRNAs, we discover that readthrough is far more pervasive than expected: we identify more than 300 readthrough events not predicted by phylogenetic approaches. We provide evidence that these novel extensions are of recent evolutionary origin, and show using specific examples that both the novel and conserved extensions can produce stable protein products, be produced in a regulated manner, and contain functional subcellular localization signals. We further demonstrate that readthrough occurs at many loci in [*psi*⁻] yeast and in primary human foreskin fibroblasts, arguing that readthrough is both a ubiquitous feature of eukaryotic translation and a novel mechanism to regulate gene expression. Stop codon readthrough thus adds plasticity to the proteome during development, and provides an evolutionary mechanism for extant genes to acquire new functions.

## Results

### Development of a ribosome profiling assay for cultured *Drosophila* cells

In order to study translation and, more specifically, stop codon readthrough in *D. melanogaster,* we sought to develop a robust ribosome profiling assay for this organism. We initially developed our protocol in S2 cells, a macrophage-like lineage derived from late-stage *Drosophila* embryos.

In previous studies, ribosome-protected fragments or 'footprints' were generated by digesting eukaryotic polysome lysates with RNase I (*Ingolia et al., 2009*, *2011*). In contrast to yeast and mammalian cell lines, we found that *Drosophila* ribosomes are highly sensitive to RNase I, potentially due to their unusual rRNA sequences and structures (*Figure 1—figure supplement 1A*; *Hancock et al., 1988*; *Jordan, 1975*; *Jordan et al., 1976*; *Pavlakis et al., 1979*). By contrast, we found that *Drosophila* ribosomes tolerate micrococcal nuclease (MNase) over a wide range of concentrations (*Figure 1—figure supplement 1B–D*). In contrast to RNase I, MNase has a strong 3′ A/T bias. This gives rise to a small amount of positional uncertainty with P-site mapping in MNase datasets, and prevents us from achieving the sort of sub-codon resolution seen in ribosome profiling datasets generated with RNase I.

Nonetheless, replicate experiments established that our measure of translation rate (the *ribosome footprint density,* defined as the number of ribosome-protected fragments per kilobase of coding region per million aligning reads in the dataset; RPKM), is highly reproducible and insensitive to changes in buffer conditions (*Figure 1—figure supplement 1E*, *Figure 1—figure supplement 2A,B*; full data in supplementary table 1 at Dryad: *Dunn et al., 2013*). Focusing on coding regions that had a minimum of 128 reads, we observed strong correlation between replicates ($r^2 = 0.998$; *Figure 1—figure supplement 2*) and an inter-replicate standard deviation of 1.07-fold, comparable to our protocols in yeast and mammalian cells. Furthermore, our measurements are robust to the number of isoforms per gene, the fraction of sequence-degenerate positions in a gene, gene length, A/T content, and distribution of ribosome density within a gene (*Figure 1—figure supplement 3*).

### Development of a ribosome profiling protocol for *Drosophila* embryos

In early (0–2 hr) *Drosophila* embryos, the vast majority of transcripts are maternally supplied and therefore regulated by post transcriptional processes, such as poly- or deadenylation, capping or de-capping, localization, degradation, and control of translation initiation. The early *Drosophila* embryo has thus been an important system for the study of post-transcriptional and specifically translational regulation (reviewed in *Lasko, 2011*).

To enable the broad analysis of these processes, we developed a sample harvesting strategy that captures the translational state of early embryos with minimal perturbation. Specifically, we developed a cryolysis protocol in which embryos are collected directly from egg-laying dishes into liquid nitrogen, homogenized while frozen, and thawed in the presence of translation inhibitors to prevent post-lysis translation. Notably, we omit dechorionation and rinsing, steps which could induce cold shock, anoxia, and related translational artifacts.

We collected replicate samples of 0–2 hr embryos, and subjected them to ribosome profiling and RNA-seq of poly(A)-selected mRNA. A subset of ribosomes partition into heavy polysomes (*Figure 1A*), consistent with reports that a distinct subset of messages is well-translated at this stage (*Qin et al., 2007*). Ribosome density measurements from replicate embryo collections are correlated nearly as well ($r^2 = 0.984$; *Figure 1B*; supplementary table 1 at Dryad: *Dunn et al., 2013*) as measurements from technical replicates from a single culture of S2 cells ($r^2 = 0.998$; *Figure 1—figure supplement 1E*). The *Drosophila* embryo thus provides a system in which experimental noise approaches the precision of our measurements, a fact that will facilitate detection of even small expression differences between wild-type and mutant fly strains.

Translational control is measured by a gene's *translation efficiency,* estimated as the ratio of ribosome footprint density (from ribosome profiling) to mRNA abundance (from mRNA-seq) for each gene. Translation efficiency measurements between replicate embryo collections are highly reproducible ($r^2 = 0.946$; *Figure 1C*) and consistent with prior measurements made by semiquantitative methods (*Figure 1—figure supplement 4*). The standard deviation of fold-changes between biological replicates is 1.19-fold (*Figure 1C*, inset), allowing detection of even modest changes in translation efficiency.

Remarkably, we find that the range of translation efficiencies for different messages spans four orders of magnitude, a range comparable to that observed for mRNA abundance of well-counted

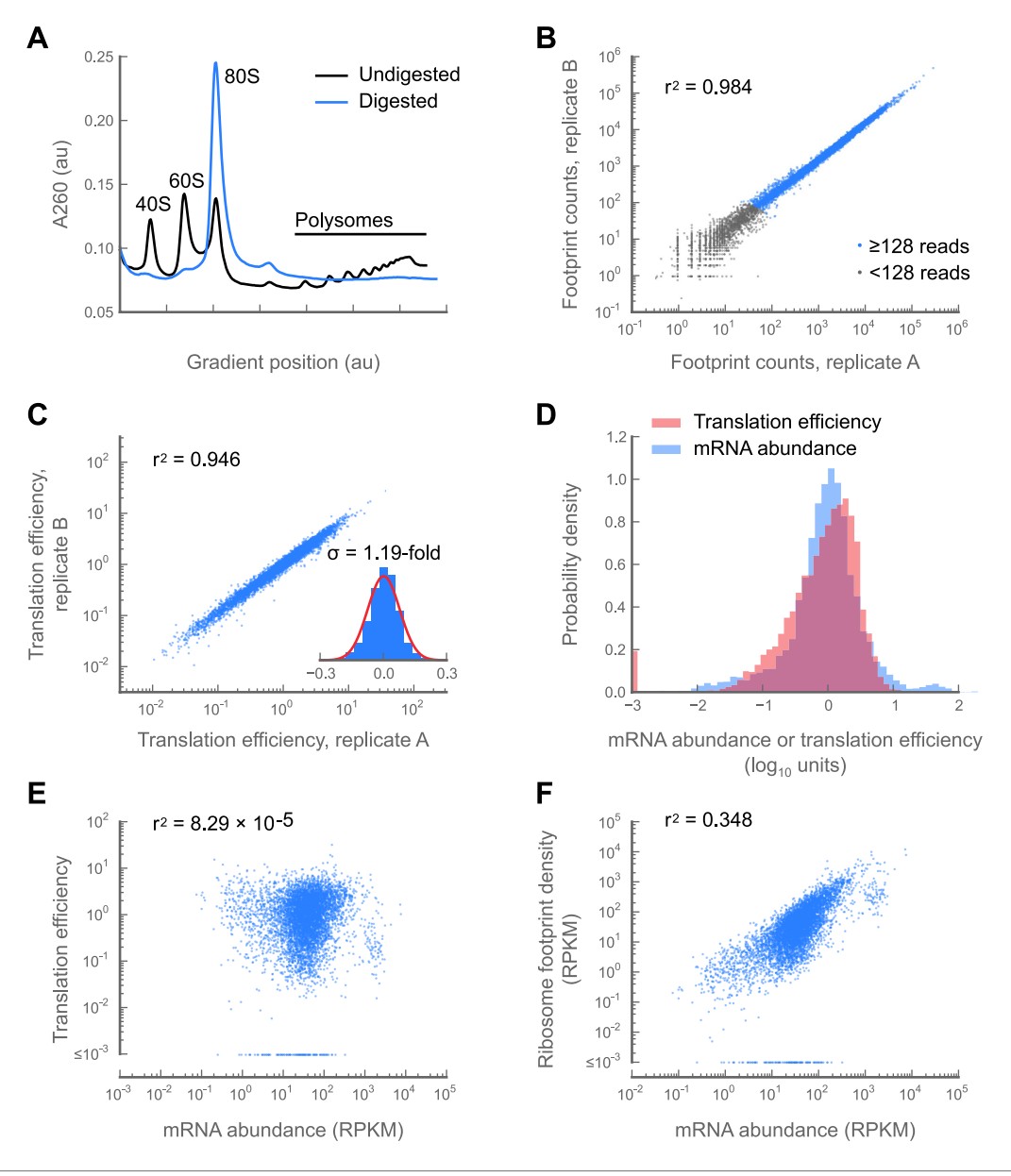

**Figure 1**. Development and validation of a ribosome profiling assay for *Drosophila melanogaster*. (**A**) Aliquots of polysome lysate from 0–2 hr embryos were fractionated on 10–50% sucrose gradients with or without prior micrococcal nuclease digestion. Digestion of exposed mRNA between ribosomes collapses the polysome peaks into the monosomal (80S) peak. The area under the monosome peak in the digested sample is 1.04-fold the combined area under the monosome and polysome peaks in the undigested sample, indicating quantitative recovery. (**B** and **C**) Measurements of translation are reproducible between replicates samples of 0–2 hr embryos. Pearson correlation coefficients ($r^2$) are shown for total ribosome-protected footprint counts in coding regions for all genes sharing at least 128 summed footprint counts between replicates (**B**), or translation efficiency measurements for all genes sharing 128 summed mRNA fragment counts between replicates (**C**). Histogram of $\log_{10}$ fold-changes in translational efficiency for each gene between two embryo replicates, along with normal error curve (**C**, inset). (**D**–**F**) Pooled data for genes containing at least 128 summed mRNA counts between both embryo replicates. Median-centered histograms of translation efficiency (pink) and mRNA abundance (blue) (**D**). Translational efficiency vs mRNA abundance for each gene (**E**). Ribosome density vs mRNA abundance for each gene (**F**). Source data may be found in supplementary table 1 (at Dryad: *Dunn et al., 2013*).

*Figure 1. Continued on next page*

*Figure 1. Continued*

The following figure supplements are available for figure 1:

**Figure supplement 1**. Digestion with micrococcal nuclease yields a robust ribosome profiling assay.

**Figure supplement 2**. Effects of buffer conditions upon reproducibility.

**Figure supplement 3**. Variability in ribosome footprint density measurements are not correlated with isoform number, sequence degeneracy in the locus of interest, locus length, A/T content, or evenness of coverage.

**Figure supplement 4**. Measurements of translation efficiency obtained via ribosome profiling are consistent with those made using semiquantitative polysome gradients.

genes (*Figure 1D*). Moreover, translation efficiency is uncorrelated with mRNA abundance ($r^2 = 8.29 \times 10^{-5}$; *Figure 1E*) and mRNA abundance predicts only one third of the variance in the rate of protein production as measured by ribosome footprint density (*Figure 1F*). Translational regulation is therefore a major determinant of gene expression in the early embryo (supplementary table 1 at Dryad: *Dunn et al., 2013*), and ribosome profiling provides a quantitative and robust means to monitor translational regulation during development.

## Ribosome density on 5′ UTRs is similar to that of coding regions

In addition to measuring gene expression, ribosome profiling maps the physical positions of ribosomes on each transcript, and thus provides a powerful tool to annotate which portions of mRNAs are translated. Consistent with our previous work in mammals (*Ingolia et al., 2011*) and yeast (*Ingolia et al., 2009*; *Brar et al., 2012*), many 5′ UTRs in *Drosophila* contain substantial footprint density (*Figure 2A*, *Figure 2—figure supplement 1*; *Supplementary file 1A*) covering sequences that appear to be upstream open reading frames (uORFs; example in *Figure 2C*).

We attribute this density to translating 80S ribosomes rather than 48S preinitiation complexes for three reasons: first, the length distribution of protected fragments in 5′ UTRs (25–35 nt) is indistinguishable from the length distribution of ribosome-protected fragments in coding regions (*Figure 2—figure supplement 2*), while the protected footprint of a preinitiation complex is reported to be larger (40–70 nt; *Lazarowitz and Robertson, 1977*; *Pisarev et al., 2008*). Second, our measurements of 5′ UTR density are indistinguishable whether we enrich digested monosomes by sedimentation through a sucrose cushion (which collects all heavy particles) or specifically separate them from preinitiation complexes by fractionation of a sucrose gradient (*Figure 2B*, *Figure 2—figure supplement 3*). Thus, the dominant signal contributing to our measurement of footprint density in 5′ UTRs is derived from fragments protected by 80S ribosomes. Third, because initiation and termination of translation are slow compared to elongation, initiation and termination events produce peaks of ribosome density (*Figure 2—figure supplement 1*; *Ingolia et al., 2011*). Such peaks are frequently visible at the boundaries of predicted uORF sequences (example in *Figure 2C*), again arguing that reads aligning to 5′ UTRs represent translation events. Given the known roles of uORFs in regulating both the translation and the stability of mRNAs (reviewed in *Meijer and Thomas, 2002*) we anticipate that our methods will facilitate future analyses of the contributions of uORFs to control of gene expression throughout fly development.

## A subset of genes exhibit stop codon readthrough, resulting in C-terminal protein extensions

Comparative analysis of the genomes of 12 sequenced *Drosophila* species has provided a powerful strategy for annotating protein-coding regions in *Drosophila* genomes (*Lin et al., 2007*, *2011*). Using this approach, 283 transcripts in *D. melanogaster* were demonstrated to contain clear phylogenetic signatures of amino acid conservation in the region between the annotated and next in-frame stop codons. It was therefore concluded that these regions encode C-terminal protein extensions (hereon called 'predicted extensions'), produced by stop codon readthrough events (*Lin et al., 2007*; *Jungreis et al., 2011*).

In our data, the density of ribosomes on 3′ UTRs is several orders of magnitude lower than in coding regions and 5′ UTRs (*Figure 2A*, *Supplementary file 1A*), and many genes show highly efficient

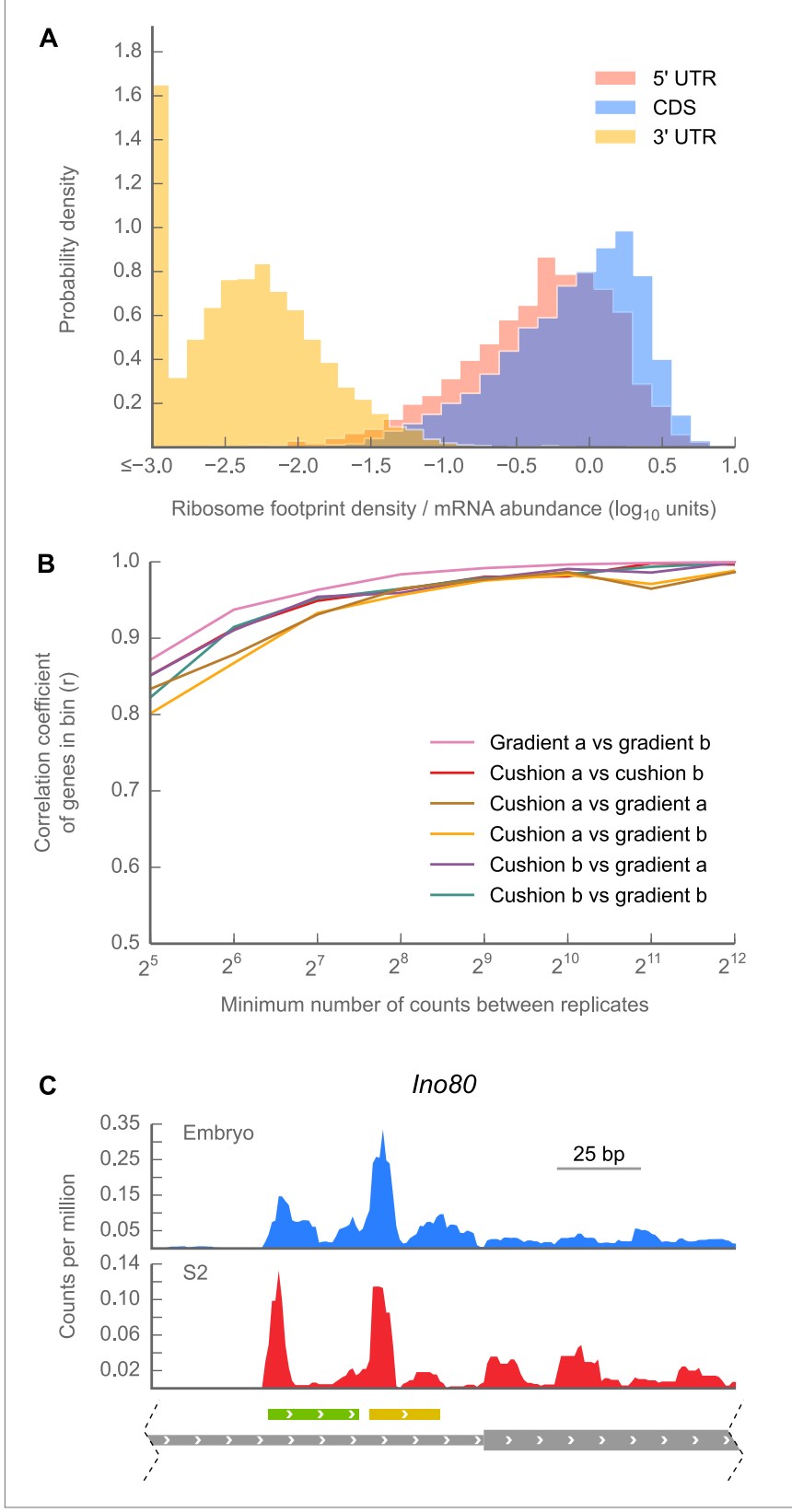

**Figure 2**. 5′ UTRs are translated. (**A**) Histograms of ribosome footprint density, corrected by mRNA abundance, for 5′ UTRs, coding regions (CDS), and 3′ UTRs in 0–2 hr embryos. (**B**) Measurements of ribosome footprint densities of 5′ UTRs agree comparably well across a range of sequencing depths, regardless of whether 80S monosomes are

*Figure 2. Continued on next page*

*Figure 2. Continued*

specifically isolated on a sucrose gradient or enriched in a cushion. For each pair of sequencing samples, Pearson correlation coefficients (r) of ribosome footprint density measurements for 5′ UTRs are plotted as a function of sequencing depth. (**C**) Example of ribosome density in 5′ UTRs corresponding to the locations of uORFs. Roughly ~200 nt of the genomic locus *Ino80* covering portions of the 5′ UTR (thin gray box) and CDS (thick gray box) are shown. In both 0–2 hr embryos and S2 cells, Initiation peaks are visible at the starts of uORFs starting with an ATG codon (green box) and a near-cognate TTG codon (yellow box) as well as at the annotated start codon (beginning of thick gray box). Source data for panels (**A**) and (**B**) may be found in supplementary table 1 (at Dryad: *Dunn et al., 2013*).

The following figure supplements are available for figure 2:

**Figure supplement 1**. Ribosome density over start and stop codons.

**Figure supplement 2**. Read lengths are similar in 5′ UTRs and coding regions.

**Figure supplement 3**. The choice of monosome enrichment technique—sedimentation through sucrose cushions or by fractionation on sucrose gradients—minimally affects of ribosome density across 5′ UTRs and coding regions. 3′ UTR measurements are noisier in samples prepared on cushions rather than gradients.

termination (example in *Figure 3B*). However, a subset of transcripts exhibit high footprint density within the predicted extensions. To determine whether the footprint density was consistent with stop codon readthrough (as opposed to alternate explanations, like frameshift), we manually scored each predicted extension whose corresponding structural gene was sufficiently expressed in our embryo sample (158 in total). An extension was scored positively if there existed ribosome density in the extension, ribosome density vanished or unambiguously decreased following the first in-frame stop codon, and positions occupied by ribosomes in the putative extension evenly covered the majority of the extension's length (see 'Materials and methods' for further details). By these criteria, 43 of the 283 transcripts predicted to undergo stop codon readthrough contained ribosome density consistent with a readthrough event (example *Figure 3C*, full data in supplementary table 2 at Dryad: *Dunn et al., 2013*), including one example of double readthrough (*Figure 3D*). We expect that the many of the remaining 240 transcripts also undergo readthrough, either at levels too low to detect at our sequencing depth, or at other developmental stages (discussed further below).

Surprisingly, we observed that a distinct set of transcripts not predicted to undergo readthrough also exhibits substantial footprint density between the annotated and next in-frame stop codons (*Figure 3E*). We therefore searched for C-terminal extensions among all transcripts that met the following criteria: (a) a minimum of 128 footprint in the corresponding CDS, (b) a minimum footprint read density of 0.2 RPKM in the extension, (c) a minimum readthrough rate of 0.001, and (d) a lack of methionine codons in the first three codons of the extension, as this latter group could be explained by initiation within the extension rather than readthrough of the upstream stop codon. We additionally excluded extensions whose translation could be explained by alternately spliced transcript isoforms that omit the stop codon. Scoring this group by the same criteria used for the predicted extensions, we identified 307 additional examples of stop codon readthrough (hereon referred to as 'novel extensions'; see example *Figure 3F*), including another example of double readthrough (*Figure 3G*). In addition, we identified several transcripts that contained 3′ UTR footprint density more consistent with ribosomal frameshift (*Figure 3—figure supplement 1A,B*), or the presence of additional downstream cistrons, RNA structure, or protein binding (*Figure 3—figure supplement 1C,D*). These were excluded from further analysis.

## Ribosome-protected footprints in C-terminal extensions show signatures of translation

Because footprint density generally is far lower in 3′ UTRs than in 5′ UTRs or coding regions (*Figure 2A*), it is possible that various sources of noise (e.g. regions of mRNA protected by RNA structures or by RNA-binding proteins) might contribute more substantially to this density than to the density in coding regions. We therefore asked whether footprints in 3′ UTRs exhibited behaviors specific to footprints protected by 80S ribosomes.

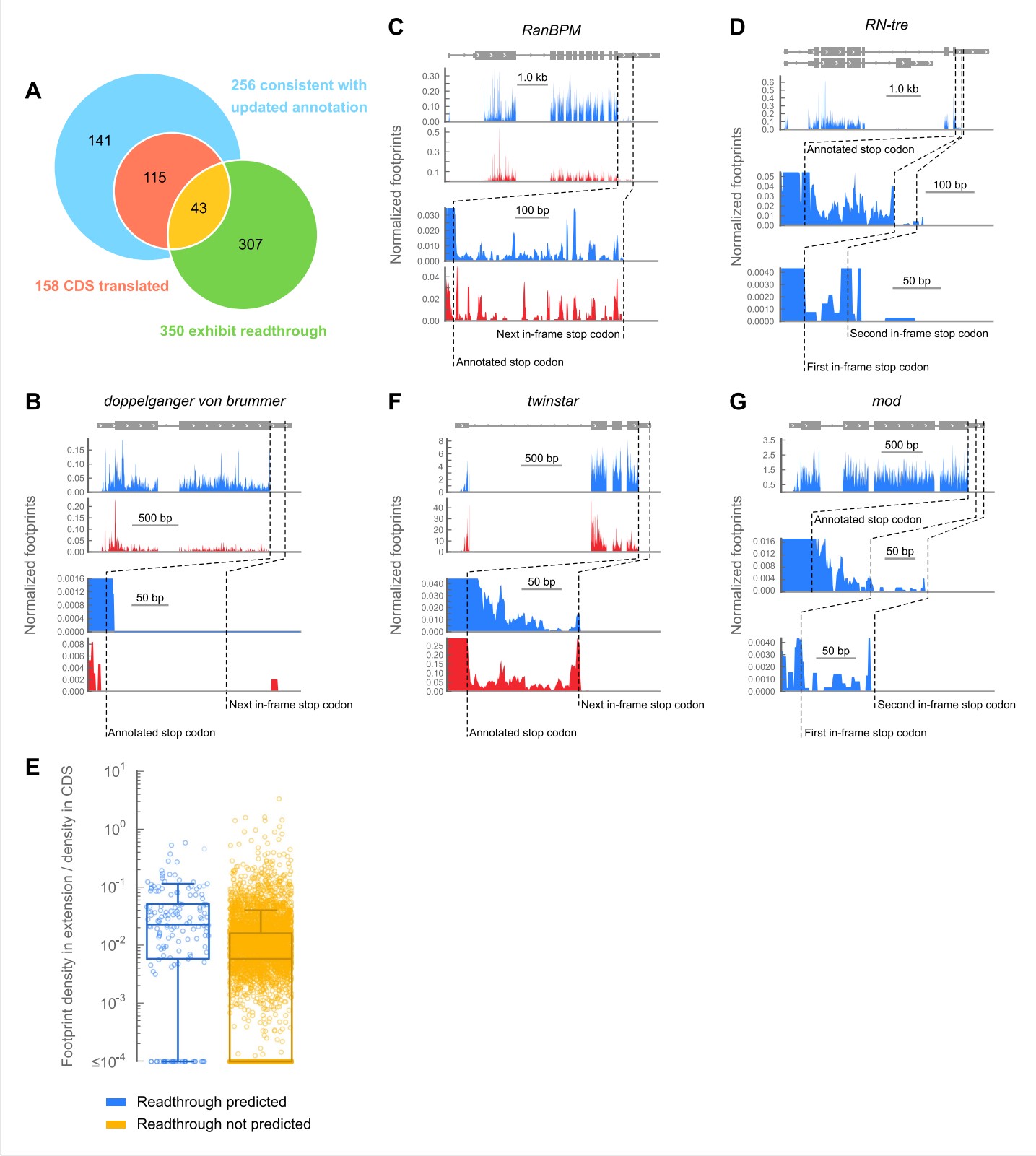

Figure 3. A subset of genes exhibit apparent stop codon readthrough. (A) Venn diagram summarizing readthrough events. Of 283 predicted extensions, 256 were consistent with FlyBase genome annotation revision 5.43. For 158 of these, the corresponding coding regions were expressed in 0–2 hr embryos. Of this subset, 43 exhibited clear signs of readthrough. Others were ambiguous, untranslated, or could be explained by other mechanisms

*Figure 3. Continued on next page*

*Figure 3. Continued*

(*Figure 3—figure supplement 1*). In addition, we identified 307 examples of readthrough that were not phylogenetically predicted. (**B**) Example of a gene that does not exhibit readthrough. Top: genomic locus with UTRs (thin boxes), introns (line), and coding regions (thick boxes). Middle: normalized footprint density covering the locus in 0–2 hr embryos (blue) and S2 cells (red) in reads per million. Bottom: magnification of region where a putative C-terminal extension would be found. Dashed lines: annotated and next in-frame stop codons (**C**) as in (**B**), except stop codon readthrough creates a C-terminal protein extension in *RanBPM*, a gene phylogenetically predicted to undergo readthrough (**D**) as in (**B**), but an example of phylogenetically predicted double-readthrough. (**E**) Ratios of the ribosome footprint density in putative extensions to corresponding coding regions. Blue: extensions predicted to undergo readthrough. Yellow: all other possible extensions. Extensions that overlapped any annotated CDS, snoRNA, or snRNA were excluded. Boxes: IQR. Whiskers: 1.5*IQR. (**F**) as in (**C**), except this transcript was not predicted to undergo readthrough. (**G**) as in (**D**), except this transcript was not predicted to undergo single or double readthrough. Source data may be found in supplementary table 2 (at Dryad: *Dunn et al., 2013*).

The following figure supplements are available for figure 3:

**Figure supplement 1**. Examples of footprint density in 3′ UTRs attributed to sources other than readthrough.

In order to distinguish whether reads mapping to extensions were either protected by ribosomes or derived from alternate sources, we compared the total number of reads aligning to extensions in samples prepared from sucrose cushions, which collect all heavy macromolecular complexes, to those in which we specifically isolated 80S ribosomes on sucrose gradients. Footprint count measurements for each extension are highly correlated between libraries made using these two sample preparation methods, indicating that these footprints are either protected by 80S ribosomes, or by another RNA binding protein that co-sediments with 80S ribosomes (*Figure 4A*; $r^2 = 0.945$).

Because various ribosome-binding proteins protect nucleotide fragments of distinct lengths, the size distribution of protected mRNA fragments provides a powerful approach for distinguishing 80S footprints from other sources (Ingolia et al., manuscript in preparation). Footprints in C-terminal extensions exhibit a length distribution very similar to footprints in coding regions, while those derived from non-coding sources, such as snoRNAs and tRNAs, show dramatically different length distributions (*Figure 4B*). Thus, footprints aligning to extensions appear to be protected by 80S ribosomes.

Finally, we sought to determine whether the ribosomes that appear to translate extensions are engaged in active translation, as opposed to some aberrant process of stalling or slippage (e.g., as described in *Skabkin et al., 2013*). Because terminating ribosomes produce a characteristic peak of ribosome density over annotated stop codons (*Figure 2—figure supplement 1*; *Ingolia et al., 2009*), we asked whether the stop codons that terminate the C-terminal extensions also showed this behavior. Indeed, C-terminal extensions exhibit peaks at their stop codons, clearly arguing that footprint density in C-terminal extensions is attributable to actively-translating ribosomes (*Figure 4C*).

Because this meta-gene analysis represents a group average, we also compiled individual statistics on ribosome release in a manner similar to the RRS score described by *Guttman et al. (2013)* . Briefly, we tabulated the ratio of the total number of reads aligning within a five codon window immediately downstream of a stop codon to the number of reads aligning to the five codon window immediately upstream of that codon, with the expectation that if ribosomes terminate at a given stop codon, the score for that codon should approach zero. We performed this calculation separately for: (1) stop codons that terminate annotated coding regions, (2) stop codons that terminate C-terminal extensions, and (3) as a negative control, randomly selected codons internal to annotated coding regions. We find that the scores of stop codons that terminate C-terminal extensions fall within the distribution of scores for stop codons that terminate annotated coding regions (*Figure 4—figure supplement 1*), again arguing that the read density covering putative C-terminal extensions are in fact produced by ribosomes that have undergone stop codon readthrough rather than other processes.

## Readthrough produces detectable translation products

It is possible that the population of ribosomes that read through stop codons is engaged in a pathological translation process that might not produce detectable protein products. We therefore asked whether we could detect translation products by immunoprecipitation (IP) and western blotting. We created reporter constructs for a panel of transcripts including five predicted and 10 novel extensions that exhibited readthrough in both 0–2 hr embryos and S2 cells. In each construct, we fused Venus (a GFP variant) upstream of a portion of each transcript containing the C-terminal 120 codons of the

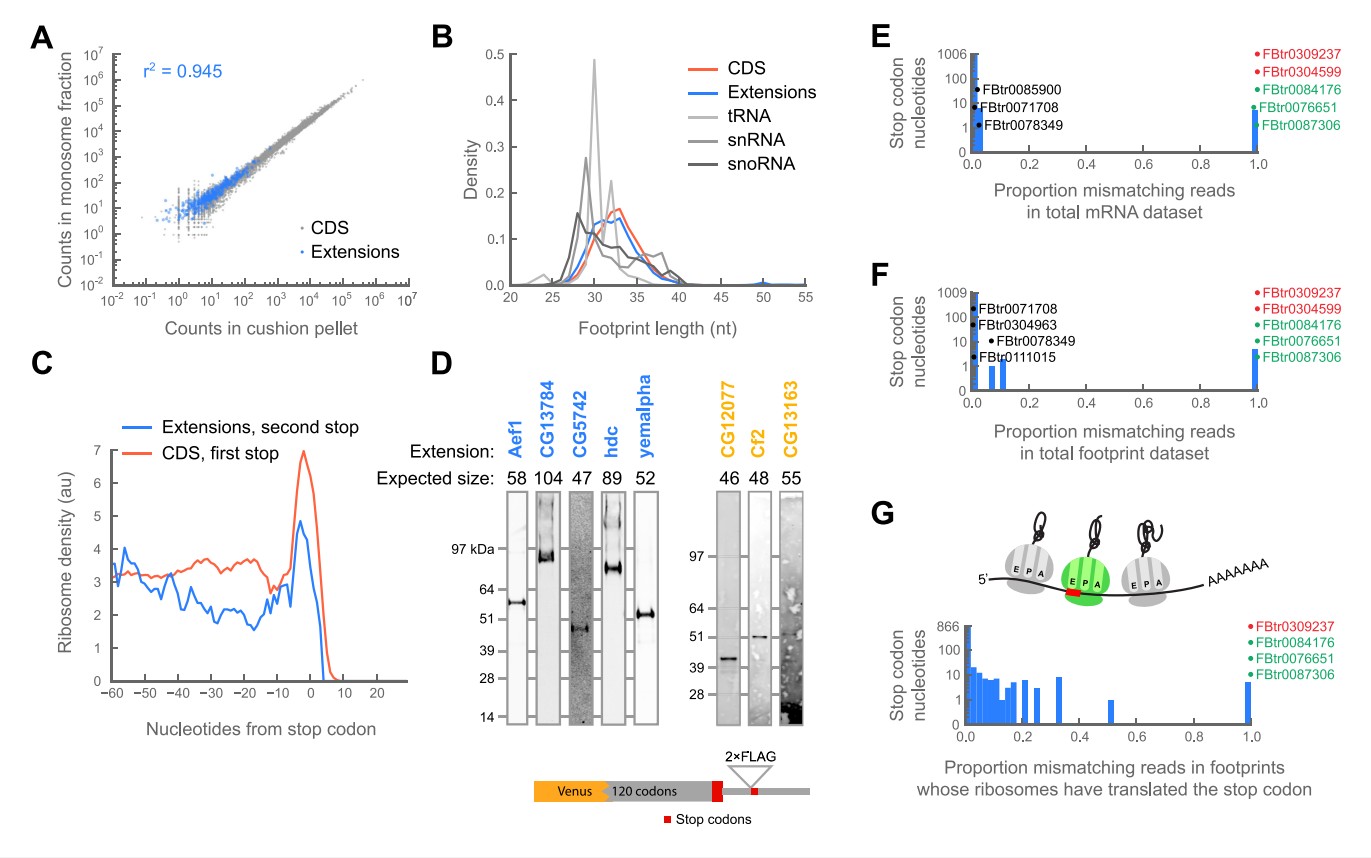

**Figure 4**. Translation downstream of the stop codon is due to readthrough. (**A**) Ribosome footprint counts for each C-terminal extension are well correlated between samples prepared by sedimentation through sucrose cushions or by fractionation on sucrose gradients (blue). For comparison, footprint counts for annotated coding regions in each sample type are plotted (gray). The Pearson correlation coefficient (r²) for C-terminal extensions is shown. (**B**) Distributions of read lengths for footprints aligning to annotated coding regions (CDS, red) and to C-terminal extensions (blue) are similar, while lengths of footprints aligning to tRNAs, snRNAs, and snoRNAs are quite different. (**C**) Meta-gene average of ribosome density at the annotated stop codons of coding regions (red), or at the stop codons that terminate extensions (blue). Both averages show characteristic peaks of ribosome density above the stop codon, characteristic of translation termination. (**D**) Readthrough produces detectable protein products. Bottom: schema of reporters. Reporters containing the GFP variant Venus fused to the 120 C-terminal codons and entire endogenous 3' UTR of a gene of interested were transfected into S2 cells. To facilitate detection of readthrough products, a double-FLAG epitope was inserted upstream of the stop codon (red) that terminates the putative extension. Top: reporters were immunoprecipitated with anti-GFP antibodies. Immunoprecipitates were then resolved by SDS-PAGE and western blotted with anti-FLAG antibodies to detect protein products of readthrough. Blue: names of genes containing extensions predicted to undergo readthrough. Yellow: names of genes containing novel extensions. (**E**) For each nucleotide in each stop codon that undergoes readthrough, we counted the fraction of reads containing nucleotide mismatches and present the data as a histogram. Transcripts containing stop codon nucleotides with significantly elevated mismatch rates are explicitly noted. Green: transcripts containing genomic polymorphisms that mutate one stop codon to another. Red: transcripts containing genomic polymorphisms that convert stop codons to sense codons. Black: other transcripts containing significantly elevated proportions of mismatches. (**F**) as in (**E**), but for ribosome-protected footprint data. (**G**) as in (**F**), but the analysis was restricted to the subset of footprints that both include the sequence of the stop codon and derive from ribosomes that have already translated the stop codon (top, green ribosome in cartoon).

The following figure supplements are available for figure 4:

**Figure supplement 1**. C-terminal extensions in *Drosophila melanogaster* show ribosome release typical of coding regions, but not of internal codons.

annotated coding sequence and the entire endogenous 3' UTR. To visualize readthrough, we fused a double FLAG epitope to the C-terminus of the putative C-terminal extension. We transfected these constructs into S2 cells, immunoprecipitated the reporter at the N-terminus using anti-GFP beads, and detected the extensions by western blotting using an anti-FLAG antibody. We detected readthrough products of the correct size for eight of the reporters, arguing that at least this subset of extensions yields C-terminally extended proteins in vivo (***Figure 4D***).

While we did not seek to detect C-terminally extended proteins generated by endogenous transcripts (e.g., through mass spectrometry), we do believe our reporter constructs to be at least as faithful as those used in earlier literature, as we included substantially more nucleotide context (120 codons upstream of stop plus the entire endogenous 3′ UTR) than other groups screening through candidate genes to find readthrough signals (2–8 codons upstream and 3–15 codons downstream of the stop codon; *Fearon et al., 1994*; *Harrell et al., 2002*; *Namy et al., 2002*, *2003*).

## Extensions are not products of selenocysteine insertion, genomic polymorphisms, or mRNA editing

The appearance of stop codon readthrough, both in ribosome profiling data and in IP-westerns, could result from several other processes, such as selenocysteine insertion, genomic mutation of stop codons to sense codons, or the editing of stop codons in mRNAs. We consider each of these in turn.

UGA stop codons may be decoded by specialized translation machinery as the unconventional amino acid selenocysteine if the 3′ UTR contains a selenocysteine insertion (SECIS) element. However, UGA stop codons represent only 25% of the readthrough events we report, and none of these are annotated as selenoproteins in either FlyBase (*Marygold et al., 2013*) or SelenoDB (*Castellano et al., 2008*). Furthermore, we were unable to detect SECIS elements in any of their 3′ UTRs using SeciSearch 2.19 (*Kryukov et al., 2003*). Thus, at most, even unannotated selenocysteine insertion events could only account for a small fraction of the readthrough events we report.

We also exclude the possibility that readthrough might result from genomic polymorphisms or RNA editing at the stop codon. Because both types of events would be represented in our data as mismatches between read alignments and the reference transcript sequence, we counted the total number of matching and mismatching reads covering each nucleotide position in each stop codon in our mRNA-seq and ribosome footprint datasets. For each dataset, we calculated a global average proportion of mismatching reads, and used the binomial test to identify stop codon nucleotides whose individual proportion of mismatches significantly deviated from the corresponding global average.

Together, the mRNA and footprint datasets identified a total of 10 nucleotide positions whose mismatch rates significantly exceeded the average (*Figure 4E,F*). Three positions contained genomic polymorphisms that changed one stop codon to another stop codon (*Figure 4E,F*, green). Two (*Figure 4E,F*, red) contained genomic polymorphisms that converted the stop codon to a sense codon. These two transcripts were therefore excluded from further study. The remaining five positions contained a variety of mismatches each occurring at low frequency. These observations are inconsistent with the presence of a genomic polymorphism at those positions, which should cause a 50% or 100% frequency of a single mismatch, depending on whether the polymorphism is hetero- or homozygous (*Figure 4E,F*, black).

An alternate explanation for a low but elevated proportion of mismatches is RNA editing, the conversion of one nucleotide to another in an mRNA. In *Drosophila*, the only mechanism known to edit mRNA is the deamination of adenine to inosine, which is converted to guanine by reverse transcriptase (*Ramaswami et al., 2013*). A-to-I editing thus appears in sequencing data as a preference for A-to-G transitions among mismatches. Of the five mismatching positions we could not ascribe to genomic polymorphisms, four contain thymine or guanine rather than adenine residues in the reference sequence, and therefore cannot be edited by this pathway. We therefore attribute these mismatches to sequencing error. The majority of mismatches at the single remaining position are transversions from adenine to thymine, similarly arguing that these mismatches are more likely due to sequencing error than to A-to-I editing.

Formally, it is possible that a minor fraction of transcripts are edited, but that this fraction, even if small as measured in the RNA-seq or total footprint data, might account for all of the stop codon readthrough we observe. Analysis of the ribosome footprint data allows us to explore this possibility directly. Specifically, were this the case, the sequences of all the footprints deriving from ribosomes that have undergone readthrough—namely, those whose A-sites have already translated the stop codon—should contain evidence of editing (*Figure 4G*, top). We therefore separately analyzed the footprints deriving from this specific pool of ribosomes. Our dataset provided sufficient coverage to test 419 of 450 such positions (93% of the total). Of these, only four stop codon positions exhibited significantly elevated levels of mismatch (*Figure 4G*, bottom). All of these were identified in the mRNA and total footprint datasets above as having genomic polymorphisms (*Figure 4E–F*). Thus, our most stringent dataset contains no positive evidence of RNA editing.

Further, this dataset contains positive evidence against RNA editing. Under the null hypothesis that A-to-I editing drives readthrough, one would expect nearly all footprints (for our purposes, conservatively assuming 90%) in the A-site footprint dataset to contain an edited base. Under this assumption, we used a binomial test to estimate the probability of observing the proportion of A-to-G mismatches in the A-site footprint dataset at each adenine residue sufficiently covered by reads (217 positions, representing roughly 50% of A positions in all readthrough events reported). In this analysis, all positions contained significantly fewer A-to-G mismatches than expected under the hypothesis of A-to-I editing, (Bonferonni-corrected $p \ll 0.05$ for all transcripts), indicating that A-to-I editing plays no part in any of the readthrough events we could test.

## Readthrough occurs in *Saccharomyces cerevisiae* and human foreskin fibroblasts

Because we detected far more readthrough events in *Drosophila* than were predicted from phylogenetic data, we collected yeast datasets and examined them for empirical evidence of readthrough. Importantly, because the [*PSI*⁺] form of the yeast eRF3 homologue is known to promote readthrough, we limited our analysis to data collected from [*psi*⁻] strains.

In contrast to MNase (which exhibits a 3′ A/T cutting bias, yielding positional uncertainty of the ribosomal P-site, see 'Materials and methods'), RNAse I shows little cutting bias. Therefore, libraries prepared with RNase I (e.g., yeast and mammalian libraries) offer superior spatial resolution along mRNAs. In such libraries, the locations of ribosome-protected footprint fragments in coding regions exhibit a characteristic three-nucleotide periodicity or *phasing* from which reading frames can be deduced (*Ingolia et al., 2009*, *2011*). We therefore tabulated the phasing of ribosome-protected footprint fragments in all annotated coding regions, putative C-terminal extensions, and the 40 codon windows downstream of the putative extensions as an approximation of the portion of the 3′ UTR distal to the putative extension (hereafter called 'distal 3′ UTRs'). To control for cloning biases caused by skewed nucleotide frequencies at each phase, we tabulated the phasing of randomly-fragmented mRNA fragments that were cloned using the same protocol and aligned to the same regions. Non-random phasing consistent with translation is apparent in both the coding regions and the putative extensions, but not the distal 3′ UTR ($p = 3.98 \times 10^{-26}$, $X^2$ test, footprints vs mRNA fragments in extension, dof = 2; *Figure 5A*). Importantly, the major component of phasing in the putative extensions occurs in the same reading frame as that of coding regions, indicating that readthrough (as opposed to, e.g., frameshift) is a major contributor to protected fragment density in 3′ UTRs in yeast. Having found global evidence for readthrough, we manually scored a subset of yeast genes to identify individual examples of readthrough, using the same filtering and scoring criteria we used in the *Drosophila* datasets. We found 30 clear examples of readthrough in yeast (examples in *Figure 5B,C*; full results in supplementary table 3 at Dryad: *Dunn et al., 2013*), demonstrating that readthrough is not unique to *Drosophila.*

Because readthrough has been observed in two mammalian genes (*Geller and Rich, 1980*; *Yamaguchi et al., 2012*), we collected data from primary human foreskin fibroblasts and sought evidence of readthrough in humans. We identified 42 readthrough events in the human data (*Figure 5D,E*; full results in supplementary table 4 at Dryad: *Dunn et al., 2013*). These events are not explained by selenocysteine insertion, and, as in *Drosophila,* read lengths mapping to extensions in the yeast and human datasets are similar to those mapping to coding regions in these organisms (*Figure 5—figure supplement 1*). Thus, readthrough appears to be prevalent in all three organisms.

To estimate how many of the novel extensions we detected might be translated at a biologically significant level, we estimated a threshold for biological significance as the fifth percentile of readthrough rates for the phylogenetically conserved extensions that were translated in the *D. melanogaster* embryo, a rate of 1.2%. Out of all the extensions for which we could measure readthrough rates (i.e., those sufficiently long not to be covered by stop codon peaks, see 'Materials and methods'), 61.8% of the novel extensions in *Drosophila,* 94.7% of the extensions in human foreskin fibroblasts, and 40.0% of the extensions in yeast exceeded this threshold, arguing that readthrough might be important in all three organisms (*Figure 5F*).

## Unpredicted C-terminal extensions show signs of recent evolutionary origin

Because 307 of the 350 readthrough events we discovered were not predicted phylogenetically, we sought to determine whether any of them showed signs of protein-coding conservation through the

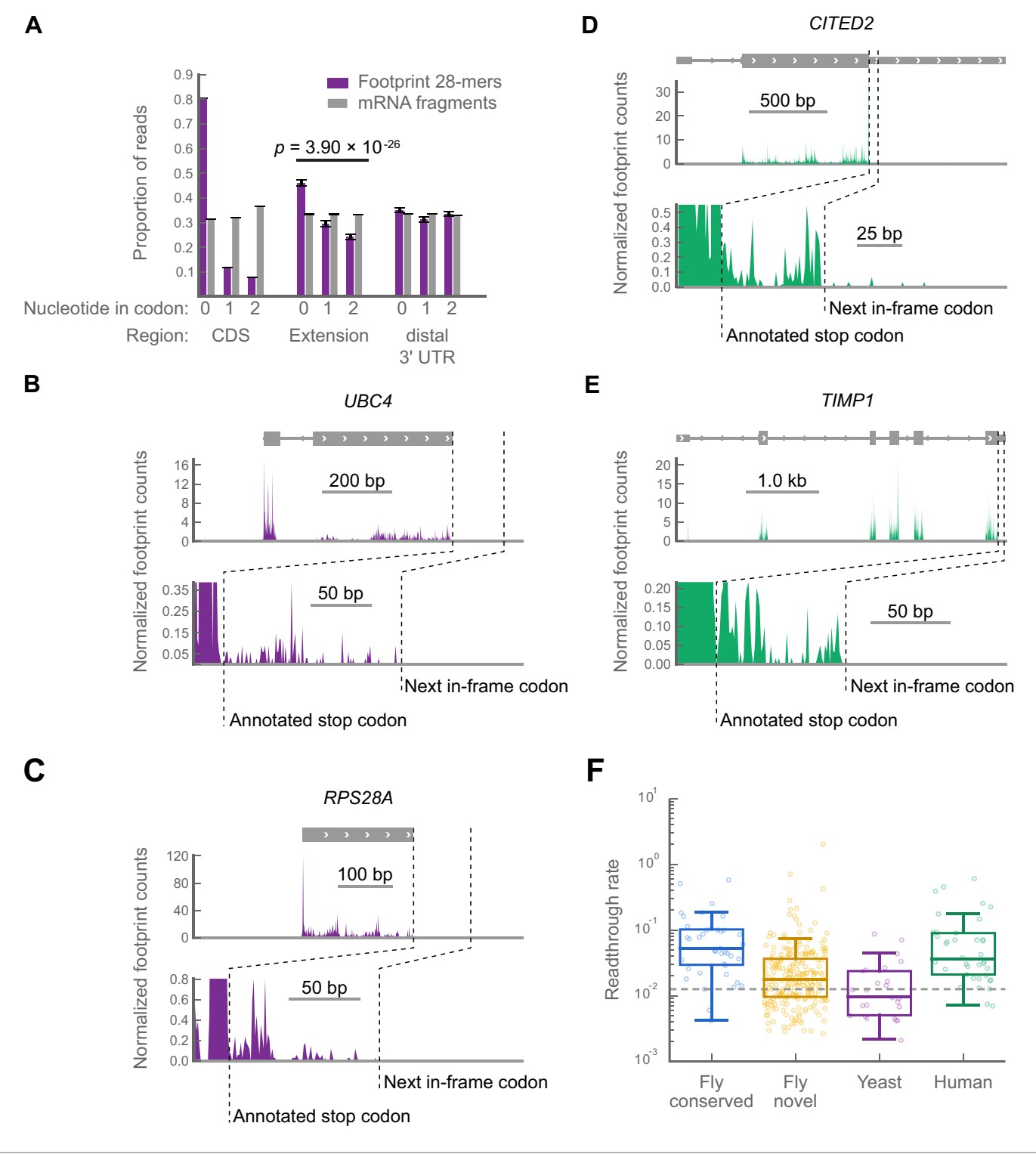

**Figure 5**. Readthrough occurs at specific stop codons in [*psi*] yeast and in human foreskin fibroblasts. (a) Triplet periodicity of 28-mers from yeast data in all non-overlapping coding regions (CDS), putative C-terminal extensions, and distal 3′ UTRs indicates that a signature of translation readthrough is visible in extensions on a bulk scale. Distal 3′ UTRs were estimated as 40 codon windows following putative extensions. Putative extensions and distal 3′ UTRs that overlap annotated coding regions, snoRNAs, snRNAs, tRNAs or 5′ UTRs were excluded from the analysis. (**B** and **C**) Examples of yeast transcripts that undergo readthrough, as in *Figure 3B*. (**D** and **E**) Examples of transcripts that undergo readthrough in human foreskin fibroblasts, as in *Figure 5. Continued on next page*

*Figure 5. Continued*

*Figure 3B*. (**F**) Distribution of readthrough rates, by organism, for all extensions of sufficient length not to be covered by bleedthrough from termination peaks ('Materials and methods'). Dashed line: fifth percentile of readthrough rate in conserved extensions in *D. melanogaster,* 1.2%. Source data may be found in supplementary tables 2, 3, and 4 (at Dryad: *Dunn et al., 2013*).

The following figure supplements are available for figure 5:

**Figure supplement 1**. In yeast and humans, reads mapping to C-terminal extensions are drawn from the same length distribution as reads mapping to coding regions.

*Drosophila* phylogeny. To this end, we used PhyloCSF, which reports a log-likelihood ratio reflecting the relative probabilities of observing a given alignment of orthologous nucleotide sequences under models of protein-coding or non-coding evolution (*Lin et al., 2011*). By this metric, only 14 of the 307 novel extensions score positively (*Figure 6A*), and their distribution of PhyloCSF scores was not markedly different from the global distribution (*Figure 6—figure supplement 1A*), indicating a lack of phylogenetic evidence for amino acid conservation.

The lack of detectable phylogenetic evidence of amino acid conservation among the novel extensions suggests two models: either (1) the novel extensions, on average, are selectively neutral, and occur only because they do not incur too great a fitness disadvantage, or (2) the novel extensions are under selection, but originated after the divergence of *D. melanogaster* from its closest sequenced relatives, making conservation in this group undetectable by cross-species tools such as PhyloCSF. To distinguish these possibilities, we used two tests to detect signs of selection for protein coding specifically within *D. melanogaster*.

To determine whether the nucleotide sequences of novel extensions show signs of selection for protein coding potential, we implemented a Z-curve classifier, a machine-learning technique that separates coding regions from non-coding regions based upon phased differences in nucleotide *k*-mer frequency (*Gao and Zhang, 2004*). We trained the classifier to distinguish annotated coding regions from distal 3′ UTRs (see 'Materials and methods' for details). Consistent with a long history of protein-coding selection, extensions predicted by phylogenetic conservation showed a nucleotide character indistinguishable from annotated coding regions (*Figure 6B*). By contrast, novel extensions exhibit a nucleotide character intermediate between coding regions and distal 3′ UTRs (p=$1.02 \times 10^{-23}$, Mann-Whitney U test, distal 3′ UTR vs novel extensions), which is consistent with an evolutionary trajectory towards coding-like character from a 3′ UTR. This effect is not due to specific nucleotide signals found in distal 3′ UTRs (p=$3.81 \times 10^{-22}$, *Figure 6—figure supplement 1B*), and was robust across Z-curve classifiers trained on different windows drawn from distal 3′ UTRs (see 'Materials and methods').

To obtain more direct evidence for or against protein-coding selection, we analyzed SNP data from 50 individuals of *D. melanogaster* from the *Drosophila* Population Genomics Project (http://www.dpgp.org). We determined the proportion of SNPs that would be synonymous if translated in-frame in coding regions, predicted extensions, novel extensions, and distal 3′ UTRs. Novel extensions show a modest but significant preference for synonymous SNPs above the background level of distal 3′ UTRs (*Figure 6C*; p=$1.42 \times 10^{-5}$, one-sided Fisher's exact test), but below that of the predicted extensions (p=$8.42 \times 10^{-9}$, one-sided Fisher's exact test). This pattern suggests that a subset of the novel extensions is undergoing selection for protein coding, and that the contribution from this subset to the average SNP preference outweighs the contributions from other subsets of extensions that are selectively neutral or undergoing diversifying selection. Together, these results favor the hypothesis that at least a fraction of the novel extensions are of recent evolutionary origin and have come under selection within the *melanogaster* lineage.

## Readthrough is regulated individually for specific transcripts

In order to determine whether C-terminal extensions might be functional, we sought evidence for biological regulation of readthrough rates. We therefore queried our S2 cell and embryo datasets for evidence of differential regulation of readthrough in all genes that were sufficiently expressed in both datasets and contain only one, unique annotated coding region across all transcripts.

For each gene meeting these criteria, we tabulated the number of ribosome-protected footprints in the corresponding coding region and extension in each tissue type, and calculated a p value for the observed distribution of counts using Fisher's exact test. Controlling the false discovery rate at 5%, we

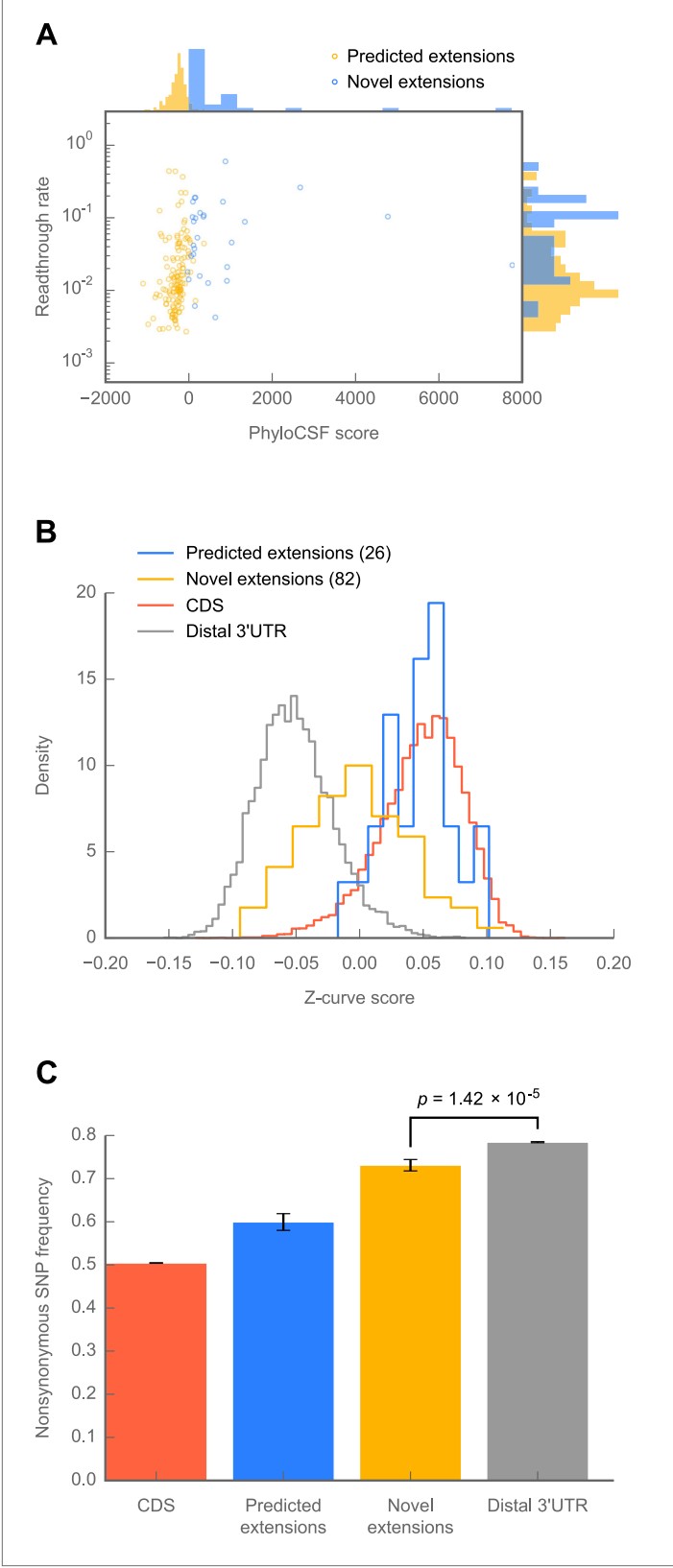

**Figure 6**. Novel C-terminal extensions in *Drosophila melanogaster* show signatures of selection within the melanogaster lineage. (**A**) Scatter plot comparing readthrough rates for confirmed extensions against PhyloCSF scores. Blue: predicted extensions. Yellow: novel extensions. Datapoints with unreliably measured PhyloCSF scores
*Figure 6. Continued on next page*

*Figure 6. Continued*

or readthrough rates are not shown ('Materials and methods'). (**B**) Z-curve classifier suggests that novel extensions have a nucleotide character intermediate between distal 3′ UTRs and coding regions. Histograms of Z-curve scores for 81-nucleotide windows drawn from annotated coding regions (CDS), distal 3′ UTRs, predicted extensions, and novel extensions. A single window was selected from each region 81 or more nucleotides long. Shorter regions were excluded from analysis, as they were empirically found to be noisy during classifier training. The Z-curve classifier was trained on windows drawn from CDS and distal 3′ UTRs as described in 'Materials and methods'. (**C**) Novel extensions accumulate SNPs with a stronger preference than distal 3′ UTRs. Proportion of SNPs in CDS, predicted extensions, novel extensions, and distal 3′ UTRs which would be nonsynonymous if translated in frame. SNPs were obtained from wild isolates of wild-type flies by the Drosophila Population Genomics Project, and were downloaded from Ensembl (*Flicek et al., 2013*). Source data may be found in supplementary table 2 (at Dryad: *Dunn et al., 2013*).

The following figure supplements are available for figure 6:

**Figure supplement 1**. Novel C-terminal extensions in *Drosophila melanogaster* show signatures of selection within the melanogaster lineage.

found nine of 182 testable transcripts to significantly change between samples, indicating that all nine should be true positives (*Table 1*; full data in supplementary table 2 at Dryad: *Dunn et al., 2013*). Thus, readthrough is differentially regulated between *Drosophila* cell types.

In principle, readthrough could be regulated by: (1) changes in the expression or activities of global factors (e.g., eukaryotic release factors, charged tRNA abundance etc), (2) by gene- or transcript-specific elements, like mRNA structures, or (3) by a combination of both. In the first scenario, readthrough rates for all transcripts should increase or decrease monotonically in one cell or tissue type compared to another. In the latter two scenarios, readthrough rates should increase for some transcripts, but decrease for others. We identified four significant increases and five significant decreases in readthrough rate in embryos compared to S2 cells, indicating that readthrough is at least in part regulated on a transcript-by-transcript basis. The distribution of fold-changes in readthrough rate spans several orders of magnitude, indicating that transcripts that are robustly read through in one cell type are not necessarily read through in another (*Table 1*). This result implies that extensions function in specific cellular or developmental contexts, consistent with earlier reports that readthrough of specific genes is regulated in metazoans (*Robinson and Cooley, 1997*; *Yamaguchi et al., 2012*).

Because we observe such a large magnitude of regulation, we believe the 350 readthrough events we report here to represent a small subset of a larger group that occur throughout the lifetime of an individual fly. We therefore expect many of the extensions that were phylogenetically predicted but not observed in our samples are in fact translated at other developmental stages in *Drosophila.* Finally, because transcripts with significant p values are statistically more highly counted in their extensions than those without significant p values (p=2.4 × 10⁻³, Mann-Whitney U test), we surmise that our ability to detect regulation was limited by sequencing depth and that the true number of transcripts whose readthrough rates are regulated in tissue- or condition-specific manners is in fact larger than we report.

## Extensions contain functional nuclear localization signals

Many peptide sequences—such as signal sequences, degrons, and phosphorylation sites—affect the localization, stability, or activity of proteins. Because these sequences are frequently short and/or degenerate, a high proportion of even random peptide sequences confer function (*Kaiser et al., 1987*; *Kaiser and Botstein, 1990*). Thus, a C-terminal extension produced by termination failure could purely by chance alter the function or behavior of its host protein, and thus come under selection. Indeed, *Freitag et al. (2012)* reported two readthrough events in fungi that append peroxisomal localization signals (PTS1) to the C-termini of glyceraldehyde-3-phosphate dehydrogenase (GAPDH) and 3-phosphoglycerate kinase, enabling these typically cytosolic enzymes to function in peroxisomal metabolism. We therefore searched our full set of C-terminal extensions for short peptide signals that direct peroxisome localization, nuclear localization (NLS), prenylation, or ER retention, or that resemble transmembrane domains (see 'Materials and methods'). PTS1 signals were detected in one extension. 10 proteins not annotated as nuclear in FlyBase contain predicted NLSes in their extensions. Eight extensions contain predicted transmembrane domains and one contains a C-terminal prenylation signal. No extension contained an ER retention signal (*Table 2*).

**Table 1.** Readthrough is differentially regulated between 0–2 hr embryos and S2 cells

| Gene ID | Alias | Embryo readthrough rate | S2 readthrough rate | PhyloCSF score | p value | $\log_{10}$ fold change | Direction of change |
|---|---|---|---|---|---|---|---|
| FBgn0036824 | CG3902 | 7.15E−01 | 2.46E−03 | −241.07 | 6.55E−10 | −2.46 | ↓ |
| FBgn0004362 | HmgD | 8.82E−03 | 1.21E−02 | −747.85 | 7.08E−07 | 0.14 | ↑ |
| FBgn0035432 | ZnT63C | 7.17E−03 | 2.71E−02 | 181.26 | 1.14E−06 | 0.58 | ↑ |
| FBgn0010409 | RpL18A | 1.39E−02 | 2.08E−03 | −197.78 | 5.85E−06 | −0.83 | ↓ |
| FBgn0039218 | Rpb10 | 5.18E−03 | 2.03E−02 | −333.38 | 8.06E−06 | 0.59 | ↑ |
| FBgn0038100 | Paip2 | 2.10E−02 | 4.60E−03 | −497.09 | 3.71E−05 | −0.66 | ↓ |
| FBgn0261790 | SmE | 7.55E−03 | 7.80E−04 | −530.28 | 9.60E−04 | −0.99 | ↓ |
| FBgn0030991 | CG7453 | 2.18E−01 | 5.28E−02 | −164.36 | 2.00E−03 | −0.62 | ↓ |
| FBgn0043796 | CG12219 | 2.85E−01 | 1.90E+00 | −27.83 | 2.11E−03 | 0.82 | ↑ |

For each transcript, the number of reads aligning to the CDS and corresponding extension were tabulated in both embryo and S2 cell datasets. p values for significant changes were calculated using Fisher's Exact Test. The False Discovery Rate was controlled at 5% using the procedures of Benjamini and Hochberg ('Materials and methods'), yielding nine transcripts with significant p values.

To determine whether any of the putative nuclear localization signals (NLSes) function in vivo, we constitutively fused C-terminal extensions containing putative NLSes to the C-terminus of a GFP-mCherry-GST reporter, which is excluded from the nucleus (*Figure 7*, left column; *Chan et al., 2007*). When expressed in S2 cells, three of four NLSes relocalized the cytosolic reporter to the nucleus at levels above background (*Figure 7*, columns 3–5), arguing that these extensions can regulate the localization of their endogenous host proteins. Given the large number of short peptide signals (e.g., phosphorylation motifs, degradation motifs, ubiquitination sequences, etc) that have been discovered, and the limited number of reporters we tested here, we likely underestimate the number of extensions that confer function. Nonetheless, our results clearly establish that C-terminal extensions can alter protein function.

## Discussion

Here we present the first comprehensive study of stop codon readthrough in a eukaryote. Using empirical data, we identified 350 readthrough events in *Drosophila melanogaster*, the vast majority of which were not predicted from phylogenetic signatures. We further demonstrate that readthrough occurs in yeast and humans. Our studies indicate that readthrough is far more pervasive than previously appreciated, is biologically regulated, and may append functional peptide signals to host proteins. Together, these results argue that stop codon readthrough provides an important mechanism to regulate gene expression and function. Our work further suggests that readthrough provides an important means for genes to acquire new functions throughout the course of evolution.

Mechanistic studies of readthrough in various systems have implicated many factors in the modulation of readthrough rates. These include the identity of the stop codon (*Robinson and Cooley, 1997*; *Chao et al., 2003*; *Napthine et al., 2012*), nucleotide context surrounding the stop codon (*Bonetti et al., 1995*; *McCaughan et al., 1995*; *Cassan and Rousset, 2001*; *Chao et al., 2003*), local or distant RNA structures (*Wills et al., 1991*; *Feng et al., 1992*; *Steneberg and Samakovlis, 2001*; *Cimino et al., 2011*; *Firth et al., 2011*; *Napthine et al., 2012*), specific hexanucleotide sequences (*Skuzeski et al., 1991*; *Harrell et al., 2002*), snoRNA-mediated pseudouridylation of stop codons (*Karijolich and Yu, 2011*), the identity of the tRNA present in the ribosomal P-site (*Mottagui-Tabar et al., 1998*), the peptide sequence of the nascent chain (*Mottagui-Tabar et al., 1998*), the concentrations of endogenous suppressor tRNAs (reviewed in *Beier and Grimm, 2001*), and proteins that bind the ribosome or mRNA (*Keeling et al., 2004*; *Hatin et al., 2007*; *Green et al., 2012*). With the exception of the readthrough signal identified in Tobacco mosaic virus (*Skuzeski et al., 1991*), the majority of readthrough events that have been mechanistically characterized are regulated by two or more such factors, often in complex, context-specific ways. For example, downstream nucleotide contexts which

**Table 2.** C-terminal extensions contain predicted functional peptide signals

| Gene ID | Alias | Extension coordinates | PhyloCSF score | Signal detected |
|---|---|---|---|---|
| FBgn0000173 | ben | X:13892649–13892781(+) | −302.18 | NLS |
| FBgn0005278 | Sam-S | 2L:113542–113647(+) | −195.30 | NLS |
| FBgn0026144 | CBP | X:7235840–7236599(+) | 128.52 | NLS |
| FBgn0031897 | CG13784 | 2L:7206347–7208015(−) | 4775.49 | NLS |
| FBgn0033712 | CG13163 | 2R:8209607–8209934(+) | −675.02 | NLS |
| FBgn0036272 | CG4300 | 3L:12265284–12265557(−) | −193.87 | NLS |
| FBgn0039213 | atl | 3R:20459429-20459720(+) | 28.43 | NLS |
| FBgn0260934 | par-1 | 2R:15370912–15371608(+) | 654.90 | NLS |
| FBgn0261606 | RpL27A | 2L:4457220–4457289^4457374–4457380(−) | −148.56 | NLS |
| FBgn0262114 | RanBPM | 2R:6322727–6323228(+) | 1045.90 | NLS |
| FBgn0031683 | CG4230 | 2L:5098384–5098573(+) | −5.34 | Transmembrane domain |
| FBgn0033712 | CG13163 | 2R:8209607–8209934(+) | −675.02 | Transmembrane domain |
| FBgn0035498 | Fit1 | 3L:4106386–4106518(+) | −323.36 | Transmembrane domain |
| FBgn0036980 | RhoBTB | 3L:20374798–20374821^20374891–20374982(+) | 154.91 | Transmembrane domain |
| FBgn0037321 | CG1172 | 3R:1221902–1222220(+) | −624.55 | Transmembrane domain |
| FBgn0040813 | Nplp2 | 3L:13350197–13350296(+) | −242.85 | Transmembrane domain |
| FBgn0053523 | CG33523 | 3L:5922386-5922854(+) | 383.85 | Transmembrane domain |
| FBgn0263864 | Ark | 2R:12913933-12914062(+) | −123.89 | Transmembrane domain |
| FBgn0039690 | CG1969 | 3R:25567115–25567154(+) | 11.52 | PTS1 |
| FBgn0035540 | Syx17 | 3L:4404848–4404983(+) | 290.83 | Farnesyltransferase signal |

Peptide sequences of C-terminal extensions were examined using various prediction servers (see 'Materials and methods'). Those containing predicted features are shown here. NLS: nuclear localization signal. PTS1: peroxisome localization signal. Coordinates are 0-indexed and half-open. Splice junctions are denoted with carrots ('^'). Strands are indicated in parentheses.

promote readthrough of one stop codon can inhibit readthrough of other stop codons, and these effects can be non-linearly synergistic with upstream nucleotide contexts (*Bonetti et al., 1995*).

Such complexity is advantageous insofar as it allows readthrough rates to be independently regulated for each transcript, consistent with our own observations. Unsurprisingly, however, this complexity has hindered efforts to identify simple *cis*-acting sequence elements that deterministically predict readthrough, and underscores the importance of having a method to measure readthrough empirically in a physiological setting in vivo. By using ribosome profiling to measure readthrough rates over a variety of tissue types and developmental stages, it may be possible to decompose the regulatory complexity into individual components, and then determine the *cis*-acting elements that collaborate to regulate readthrough in tissue-specific manners.

Just as alternative splicing provides a means for proteins to acquire new domains or functional modules, we propose, along with the Lindquist (*True and Lindquist, 2000*) and Kellis (*Jungreis et al., 2011*) groups, that stop codon readthrough can provide a mechanism for proteins to evolve at the C-terminus. In this model, transcripts that contain contexts favorable to leaky termination would yield substoichiometric, C-terminally extended populations of cellular proteins. If a particular extension is deleterious, natural selection can favor mutations in the corresponding mRNA that promote efficient

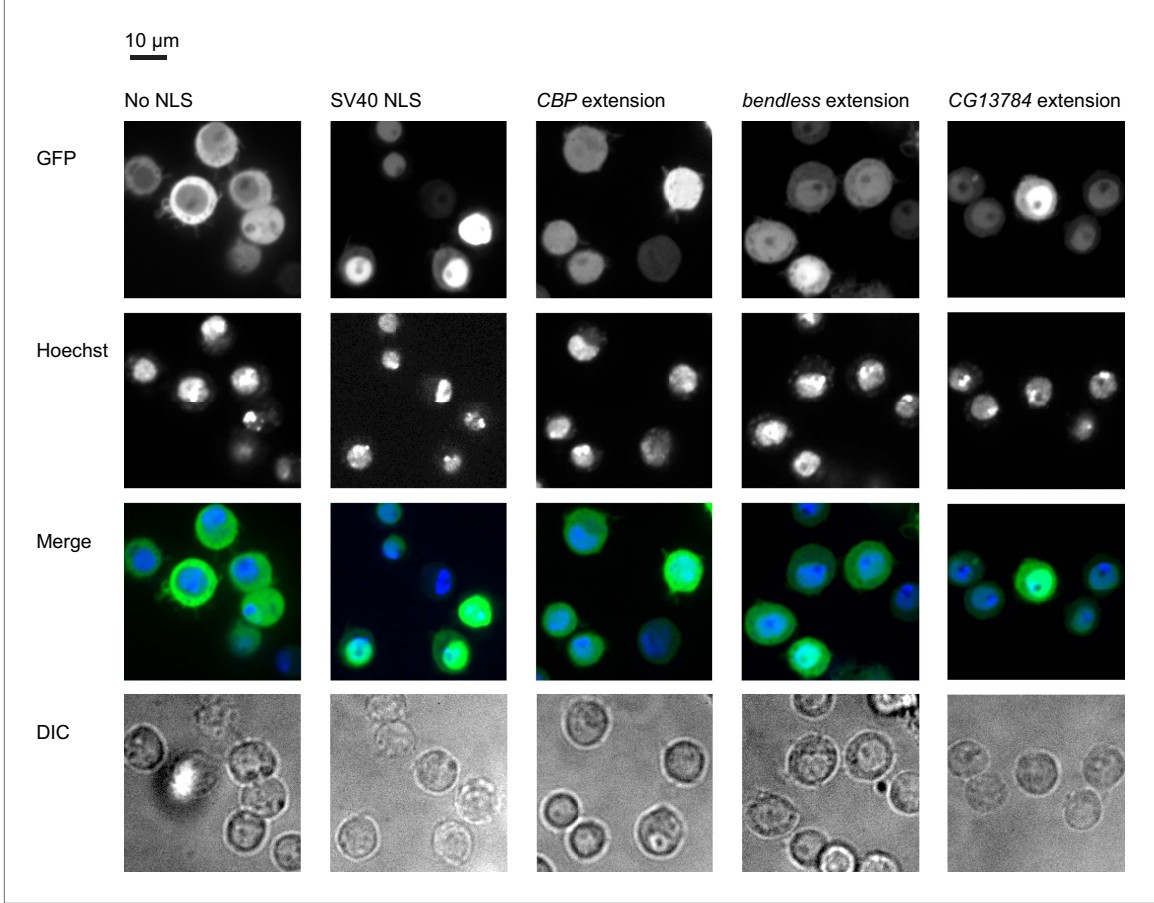

**Figure 7**. Extensions contain functional localization signals. Ordinarily, a GFP-mCherry-GST reporter is excluded from the nucleus (first column). When an SV40 NLS is appended to the reporter, it is predominantly nuclear (second column). Three extensions also contain functional NLSes which at least partially relocalize the reporter to the nucleus when constitutively fused to it (remaining columns). First row: GFP reporter. Second row: nuclei stained with Hoechst. Third row: merged GFP and Hoechst. Fourth row: DIC.

termination rather than readthrough. If, instead, the extension provides a fitness advantage, selection can act upon both its amino acid sequence (to tune its function), as well as the nucleotide sequence of its mRNA (to increase or otherwise regulate its readthrough rate). In extreme cases, where an extension is universally advantageous, a mutation that changes a stop codon to a sense codon might become fixed, resulting in a constitutively extended gene. Conceivably, the two C-terminal extensions that we discovered to contain sense codons in place of their annotated stop codons could be the end result of this process.

Several lines of evidence are consistent with this evolutionary model. First, non-zero readthrough rates (0.02–1.4%) have been observed even for control non-readthrough reporter constructs in [*psi⁻*] yeast (*Fearon et al., 1994*; *Bonetti et al., 1995*; *Namy et al., 2002*; *Keeling et al., 2004*; *Torabi and Kruglyak, 2011*) and mammalian cells (*Firth et al., 2011*; *Napthine et al., 2012*), arguing that under typical conditions in a variety of eukaryotes, there is a small pool of C-terminally-extended proteins, originating from a wide variety of genes, available for selection to act upon.

Secondly, in specific circumstances, selection appears to favor leaky termination and its extension products. Torabi et al. (*Torabi and Kruglyak, 2011*) reported that in a panel of wild strains of [*psi⁻*] yeast, allelic combinations of *SUP45* and *TRM10* that promote and inhibit readthrough appear to be in balancing selection, implying that a low baseline level of readthrough is beneficial. Similarly, numerous reports have demonstrated that wild strains of yeast exhibit [*PSI⁺*]-dependent fitness advantages in a variety of stress conditions, arguing that functions conferred by C-terminal extensions can provide adaptive advantages (*True and Lindquist, 2000*; *Halfmann et al., 2012*).

Thirdly, extensions have a high probability of conferring function without prior tuning by natural selection. This point is illustrated by the studies of Kaiser and Botstein, which demonstrated that a large proportion—roughly 30%—of randomly-generated peptide sequences are functional, insofar as they can relocalize a cytosolic form of invertase to the nucleus, mitochondrion, or endoplasmic reticulum in yeast (**Kaiser et al., 1987**; **Kaiser and Botstein, 1990**). Given the large number of short peptide signals now known (e.g., D-boxes, KEN-boxes, SH3 binding epitopes, phosphorylation sites, etc), it is likely that a far greater fraction of random peptide sequences contain at least one functional signal. Consistent with this hypothesis, we discovered C-terminal extensions that are not phylogenetically conserved nonetheless contained functional NLSes in *Drosophila.* Furthermore, because these short signals are modular, their addition to the C-terminus of a protein can confer novel function, without requiring modification or coevolution of the host protein. In this way, even novel C-terminal extensions arising purely from termination failure can immediately alter the behavior of their host proteins, in beneficial or deleterious ways, and thus come under selection. Over evolutionary time, this process could yield phylogenetically conserved readthrough events or, in extreme cases, constitutively extended proteins.

Our model predicts that, at any given moment, ribosome profiling should detect a broad spectrum of conservation among readthrough events: at one extreme are ancient, phylogenetically conserved extensions, and, at the other, extensions of recent evolutionary origin. Between these, one would find extensions under varying degrees of age, conservation, and selection. This notion is borne out in our data: in *Drosophila,* a subset of readthrough events are well supported by conservative codon substitutions across the phylogeny (**Lin et al., 2007**; **Jungreis et al., 2011**), but a far larger set is not conserved between species. In aggregate, this non-conserved group shows weak but statistically significant signals of selection among fifty wild-type individuals of *D. melanogaster* (**Figure 6C**), suggesting that a fraction of this group is undergoing protein coding selection. The remainder might include many other different groups of extensions: namely, a group of extensions undergoing diversifying selection, a group of deleterious extensions undergoing counterselection, and a group of selectively-neutral extensions subject to genetic drift.

Finally, our model predicts that conserved extensions should on average exhibit higher readthrough rates than novel extensions, because only a subset of the latter group would have been selected for function and regulation. Our data is also consistent with this prediction: the median readthrough rate for the conserved extensions in *Drosophila* is 5.2%, while the median for the novel extensions is 1.7%. Notably, 62% of the novel extensions we identified in *Drosophila,* 95% of the extensions in human foreskin fibroblasts, and 40% of the extensions in yeast undergo readthrough at rates comparable to those of phylogenetically conserved extensions (**Figure 5F**), arguing for the importance of these subsets.

Broadly, our work builds upon the growing amount of evidence that eukaryotic genomes and proteomes are far more plastic than previously thought, particularly with regard to translation and coding. In addition to the large number C-terminal extensions we report, various groups have used ribosome profiling to determine that large numbers of genes are regulated by uORFs that initiate at near-cognate start codons (**Ingolia et al., 2009**; **Brar et al., 2012**), that many genes can be N-terminally extended in a regulated manner (**Fritsch et al., 2012**), and even that many parts of mammalian genomes are decoded in multiple frames (**Michel et al., 2012**). Given the preeminence of *Drosophila* as a developmental model and the abundance of conditional genetic tools available, we anticipate that ribosome profiling in *Drosophila* will be useful in deciphering the biological roles of not only readthrough, but all non-canonical translation events, throughout development.

## Materials and methods

### Cell culture

Wild-type (*y w*) flies were cultured according to standard procedures. S2 cells were cultured in Schneider's (Gibco by Life Technologies, Carlsbad, California) media supplemented with 10% heat-inactivated FBS (UCSF cell culture facility, San Francisco, California) and antibiotics (UCSF cell culture facility). S2 cells were transfected using Effectene reagent (Qiagen, the Netherlands) following the manufacturer's instructions. For stable transfectants, the plasmid of interest was co-transfected at a 10:1 molar excess with pCoPuro. Stable integrants were selected and maintained in Schneider's media supplemented as above, but additionally containing 10 µg/ml puromycin.

## Lysate preparation

### S2 cells

16–20 hr before an experiment, cultures were diluted to 1.5–1.8 million cells/ml. To start the experiment, cells were treated for 2 min with 0.01 vol 2 mg/ml emetine (Sigma-Aldrich, St Louis, Missouri), pelleted for 2 min at 1600 rpm in a tabletop centrifuge, resuspended in 4–6 cell volumes cold polysome lysis buffer (50 mM Tris pH 7.5, 150 mM NaCl, 5 mM MgCl$_2$, 0.5% Triton x-100, 1 mM DTT, 20 U/ml Superaseln (Ambion by Life Technologies), 20 µg/ml emetine), and homogenized on ice in a pre-chilled dounce homogenizer. The resulting lysate was clarified by spinning 10 min at 20,000 × $g$ at 4°C in a microcentrifuge. Clarified lysate was aliquoted, flash-frozen in liquid nitrogen, and stored at −80°C. Experiments used 12–96 ml S2 cell culture, depending on the application. For ribosome profiling, a single 12 ml culture is sufficient.

### Embryos

0–2 hr old wild-type ($y$ $w$) embryos were collected from egg laying dishes directly into a 50 ml conical tube full of liquid nitrogen using a rubber policeman. Multiple collections were pooled until roughly 200 µl embryos had been collected for each sample. The liquid nitrogen was then decanted, the tube capped, and the pooled embryos stored at −80°C. Frozen pellets of a modified polysome lysis buffer additionally including 50 µM GMP-PNP (Sigma-Aldrich) were prepared by dripping buffer into a conical tube of liquid nitrogen. The nitrogen was decanted, the tube capped, and buffer pellets stored at −80°C. Frozen embryos and 4–6 vol of frozen buffer pellets were ground together 6 times for 2 min each at 15 Hz in a TissueLyser (Qiagen), chilling the canisters in liquid nitrogen before and after each round of grinding. Grindate was either stored at −80°C, or thawed immediately under running tepid water. Thawed grindate was clarified by spinning at 3,000 × $g$ in a tabletop centrifuge. Avoiding the wax and fat layers at the top, the supernatant was collected into pre-chilled microcentrifuge tubes, and clarified again by spinning 10 min at 20,000 × $g$ at 4°C. Lysates were aliquoted, flash-frozen in liquid nitrogen, and stored at −80°C.

## Ribosome footprinting

Concentrations of total RNA in lysates were determined using the RiboGreen kit (Molecular Probes by Life Technologies). For each sample, 35–100 µg total RNA was diluted 2:1 in digestion buffer (50 mM Tris pH 7.5, 5 mM MgCl2, 0.5% Triton x-100, 1 mM DTT, 20 U/ml Superaseln, 20 µg/ml emetine, 15 mM CaCl$_2$, and 3 U micrococcal nuclease [Roche Applied Science, Indianapolis, Indiana] per µg of total RNA in the sample), to bring the final concentration of NaCl to 100 mM and CaCl$_2$ to 5 mM. Samples were digested for 40 min at 25°C in a Thermomixer (Eppendorf, Hamburg, Germany). Digestions were quenched by adding EGTA to a final concentration of 6.25 mM and placing the reactions on ice. 1 U MNase is defined as previously (*Oh et al., 2011*) as an increase of 0.005 A260 per min, measured in a Spectramax M2 plate reader (Molecular Devices, Sunnyvale, California) using 10 µg/ml salmon sperm DNA (Sigma-Aldrich) with 5 mM Ca+ and 20 mM Tris, pH 8.0 in a 0.1 ml reaction at 25°C.

## Sucrose gradients

10–50% sucrose gradients were prepared in polysome gradient buffer (250 mM NaCl, 15 mM MgCl$_2$, 20 U/ml Superaseln, 20 µg/ml emetine) using a GradientMaster (Biocomp Instruments, Fredericton, New Brunswick, Canada) in polyclear centrifuge tubes (Seton Scientific, Petaluma, California). Up to 200 µl of samples was applied to the top of each gradient. Gradients were resolved by spinning for 3 hr at 35 krpm at 4°C in an SW-41 rotor (Beckmann Coulter, Brea, California), and fractionated using the GradientMaster. When appropriate, monosome fractions were collected, flash-frozen in liquid nitrogen, and stored at −80°C.

## Sucrose cushions

Up to 0.5 ml of digested sample was layered atop 1.0 ml of a solution of 34% sucrose in polysome gradient buffer. Monosomes were sedimented by spinning for 4 hr at 70 krpm at 4°C in a TLA-110 rotor (Beckmann Coulter). Pellets were resuspended in 600 µl 10 mM Tris, pH 7.0 and stored at −20°C.

## Ribosome profiling of *D. melanogaster*

Lysates were prepared and footprinted as above. Unless otherwise indicated, monosomes were enriched by sedimentation through 34% sucrose cushions and resuspended in 600 µl 10 mM Tris,

pH 7.0. Resuspended monosomes were extracted once with 700 µl 65°C acid phenol and 40 µl 10% SDS, followed by 650 µl acid phenol and a final extraction with chloroform. RNA was precipitated for at least 2 hr at −30°C, resuspended in 10 mM Tris, pH 7.0, and quantitated on a NanoDrop spectrophotometer (Thermo Scientific, Asheville, North Carolina). 5–35 µg RNA was dephosphorylated for 1 hr at 37°C using T4 polynucleotide kinase (New England Biolabs, Ipswich, Massachusetts) in a 50 µl reaction and resolved on a 15% TBE-urea gel (Invitrogen by Life Technologies). A gel slab spanning 28–34 nt (as measured by oligoribonucleotide size standards in a neighboring lane; see *Supplementary file 2*) was excised from the gel, eluted, and precipitated. Samples were then carried through all steps of library generation (see below).

## Poly(A)+ RNA-seq of *D. melanogaster*

For each sample, 375 µl of undigested polysome lysate was diluted into 3 vol Trizol LS (Invitrogen) and total RNA was extracted following the manufacturer's instructions. 20–50 µg Poly(A)+ RNA was selected on oligo-dT25 DynaBeads (Invitrogen) per manufacturer's instructions, and fragmented at 95°C in fragmentation buffer (2 mM EDTA, 100 mM $NaCO_3$/$NaHCO_3$, pH 9.2) to a mean size of roughly 100 nt. Fragmented RNA was precipitated, dephosphorylated for 1 hr at 37°C with T4 polynucleotide kinase (New England Biolabs), and resolved on a 15% TBE-urea gel. A gel slab corresponding to 55–65 nt was excised from the gel, eluted, and precipitated. Samples were then carried through all steps of library generation (see below).

## Subtractive hybridization to remove rRNA-derived fragments

We performed two sequential rounds of subtractive hybridization on each sample. To 5 µl cDNA the following were added: 1 µl 20× SSC, 3 µl nuclease-free water, and 1 µl of a 60 µM mixture of the biotinylated oligonucleotides oJGD132, oJGD133, oJGD134, oJGD135, oJGD136, oJGD161, oJGD162, oJGD163, and oJGD164 (sequences in *Supplementary file 2*) mixed in a ratio of 25.5:1:13:17:4:6:2:11:21. Samples were denatured for 90 s at 95°C and annealed for 20 min at 25°C. MyOne Streptavidin C1 DynaBeads (Invitrogen) were prepared as follows: for each sample, 45 µl of beads were aliquoted into a microcentrifuge tube and washed three times in 50 µl 2 × binding buffer (10 mM Tris, pH 7.5, 1 mM EDTA, 2 M NaCl), and resuspended in 22.5 µl 2 × binding buffer. 10 µl equilibrated beads were added to 10 µl hybridized sample. The mixture was incubated at 20 min in a room temperature Thermomixer with shaking at 850 rpm. Beads were then separated on a magnetic manifold (Invitrogen) and the supernatant recovered to a microcentrifuge tube.

For the second round of subtraction, 1 µl 60 µM biotinylated oligo mix and 1 µl 20X SSC were added to the supernatant from the first subtraction, and the denaturation and annealing repeated. 10 µl of equilibrated beads were pelleted on a magnetic manifold. The buffer was removed, and the beads resuspended in the mixture from the second hybridization. Samples were then incubated at 20 min in a room temperature Thermomixer with shaking at 850 rpm. The supernatant was recovered on a magnetic manifold, transferred to a microcentrifuge tube, precipitated, and resuspended in 15 µl 10 mM Tris, pH 8.0.

## Library generation

RNA concentrations were measured using the Small RNA Series II Bioanalyzer assay (Agilent Technologies, Santa Clara, California). 10–15 picomoles of RNAs were ligated to 1 µg 3′ miRNA cloning linker 1 (Integrated DNA Technologies, Coralvaille, Iowa) for 2 hr 30 min at 25°C in ligase buffer (1× T4 RNA ligase 2 buffer [New England Biolabs], 40% PEG-100 [Sigma-Aldrich], 5% DMSO, T4 RNA ligase 2 K227Q, truncated [a kind gift from Calvin Jan]) in a 20 µl reaction. Ligated fragments were precipitated for at least 2 hr at −30°C, purified on a 10% TBE-urea gel, eluted, and precipitated. Ligation products were then reverse-transcribed using SuperScript III (Invitrogen) in a 16.7 µl reaction using using the primer o225-link1 (see *Supplementary file 2*). RNA template was hydrolyzed by addition of 1/10 vol 1 M NaOH and incubation at 95°C for 20 min cDNAs were purified on a 10% TBE-urea gel (Invitrogen), eluted, precipitated, and resuspended in 5 µl 10 mM Tris pH 7.0. cDNAs from footprint samples were subjected to two rounds of subtractive hybridization as described above.

Subtracted samples were circularized using CircLigase (Epicentre, Madison, Wisconsin), following manufacturer's instructions in a 20 µl reaction. An additional microliter of CircLigase was then added, and the circularization repeated a second time. Circularized libraries were amplified by 6–12 cycles of PCR using oNTI231 and any of four indexing primers oCJ30–33 (*Supplementary file 2*) using Phusion

polymerase (Finnzymes by ThermoScientific) in a 17 µl reaction. Amplification products were size-selected on 8% TBE gels (Invitrogen), eluted, precipitated, and resuspended in 10 µl 10 mM Tris, pH 8.0. Samples were then quantitated using the Bioanalyzer High Sensitivity DNA assay (Agilent Technologies), diluted to 2 nM, multiplexed as needed, and subjected to 50–57 cycles of single-end sequencing on an Illumina HiSeq sequencer (Illumina, San Diego, CA) using version 3 clustering and sequencing kits with a 6-cycle index read (Illumina).

## Sequence processing and alignment

For all *Drosophila* experiments we used revision 5.43 of the FlyBase genome annotation and the corresponding genome assembly (*Marygold et al., 2013*). Reads were demultiplexed and cleaned of 3′ cloning adapters using in-house scripts. Reads shorter than 25 nt were discarded. Remaining reads were aligned using Bowtie version 0.12.7 (*Langmead et al., 2009*) sequentially to Bowtie indices composed of the following sequences: (a) *D. melanogaster* rRNAs (GenBank accession #M21017 [*Tautz et al., 1988*] and from FlyBase), (b) *D. melanogaster* tRNAs, snoRNAs, and snRNAs (from FlyBase), (c) cloning oligos, (d) the S288C yeast genome version R64-1-1 (downloaded on 6 June 2011 from http://downloads.yeastgenome.org/sequence/S288C_reference/genome_releases/), (e) *Wolbachia* (GenBank accession #AE017196), (f) *D. melanogaster* chromosome arms, and (g) splice junctions (from FlyBase and, in the case of embryos—figure supplemented with junctions discovered in the pooled embryo mRNA datasets using HMMSplicer 0.95 [*Dimon et al., 2010*]). For all quantitative analyses, we counted only uniquely-mapped reads.

Alignments were assigned to genomic coordinates as follows. Randomly-fragmented poly(A)+ mRNA alignments were counted along the entire length of the alignment. Each genomic position covered by a single RNA fragment was incremented $1/l$, where $l$ corresponds to the length of the alignment. Ribosome-protected footprint alignments were mapped to their estimated P-sites as follows: 12 nt were pruned from each end of the alignment, leaving a fragment $n$ nt long (where $n = l - 2 \times 12$). Each genomic position covered by a nucleotide remaining in the pruned alignment was then incremented by $1/n$. Thus, the P-site of each 25 mer was assigned to one unique position, while the P-site of each 26-mer was spread over two positions, each incremented by 0.5 reads, and so on. Alignment statistics are given in *Supplementary file 1B*.

## Attribution of counts to genes and transcripts

### Masking of degenerate genomic positions

To determine which positions in the genome give rise to reads that fail to uniquely map, we divided the genome into all possible 29-mers centered on each nucleotide position, and aligned the resulting 29-mer back to the genome allowing zero mismatches. If the 29-mer aligned to multiple sites, the position from which it arose was flagged as degenerate. All such positions were excluded from further analysis.

### Attribution of nucleotide positions to loci

Because the genome annotation contains polycistronic transcripts in which each cistron is annotated as belonging to a separate gene—for example *tarsal-less/polished rice*, which is annotated as four separate genes (FBgn0259730–3)—we collapsed each set of genes whose transcripts share exons (370 genes total) into 179 merged loci. All nucleotide positions in any transcript deriving from a locus were attributed to that locus. Any nucleotide position attributed to multiple loci (e.g., overlapping genes on the same strand), were excluded from further analyses on the gene or transcript levels.

### Attribution of nucleotide positions to exons, 5′UTRs, 3′ UTRs, and coding regions

For each locus, any position included in any transcript deriving from that locus was included in the list of exonic positions for that gene. Any exonic position which could be labeled as two or more of CDS, 5′UTR, or 3′ UTR depending on the transcript isoform was still counted as exonic, but was excluded from analyses that required positions to be uniquely labeled (e.g., comparisons of translation in 5′ or 3′ UTRs to CDS) unless otherwise noted.

### Filtering of countable loci

For all analyses, we counted only loci or transcripts deriving from loci that contain at least 95% non-degenerate positions and are at least 60 nucleotides in exonic length, after exclusion of degenerate positions and positions covered by multiple loci. Genes and transcripts that are not translated but

which may contaminate the data due to their abundance (*e.g.,* those that encode microRNAs, rRNAs, snRNAs, snoRNAs, and tRNAs) were excluded from analysis. We also excluded the loci *mod(mdg-4)* (which contains transcripts deriving from both strands) and *Yeti,* (for which transcript annotations existed on chromosome arms 3R and 2RHet).

## Measurements of gene expression

mRNA abundance and ribosome density for each genomic feature were measured in reads per kilobase of feature length per million reads aligning to chromosomes or splice junctions in the dataset (RPKM), a unit which corrects for both feature length and sequencing depth. Unless otherwise indicated, the RPKM values we report for mRNA abundance reflect the total number of RNA fragments aligning to all countable exonic positions for a given locus. For ribosome density, we report the total number of ribosome-protected footprint fragments aligning to all countable positions of a coding region (CDS) for a given locus. We calculate translation efficiency as the ratio of footprint RPKM in the CDS to the RNA fragment RPKM across the entire locus. When comparing mRNA fragment or footprint density between samples, we restricted our analyses to genes that had at least 128 summed counts between replicates as determined in *Figure 1—figure supplement 2*. When comparing translation efficiencies between samples, we required at least 128 exonic counts of mRNA for each gene.

## Translation efficiency of 5′ UTRs, CDS, and 3′ UTRs

Translation efficiencies for these regions were calculated as the ratio of footprint counts to mRNA counts in each region, for all regions with at least 128 mRNA counts. We excluded all positions that could be labeled as two or more of 5′ UTR, CDS, or 3′ UTR depending upon transcript isoform. To remove variability or bleedthrough introduced by start and stop codon peaks, we additionally excluded the following genomic positions from consideration: 9 nucleotides preceding each start codon, 15 nucleotides following each start codon, the 15 nucleotides preceding each stop codon, and the 15 nucleotides following each stop codon.

## Identification of C-terminal protein extensions

### Mapping predicted extensions to transcripts in the modern annotation

C-terminal extensions predicted by *Jungreis et al. (2011)* were mapped onto the FlyBase annotation 5.43 as follows: First, 26 predicted extensions that overlap regions that are annotated as coding (for reasons other than readthrough) in the present annotation were excluded from further analysis. One additional extension was excluded because it overlapped the 5′ UTR of another gene. The remaining 256 extensions were mapped to transcripts in FlyBase r5.43 that satisfied the following criteria: (a) if the transcript contains an annotated 3′ UTR, it fully covers the extension and (b) the transcript's annotated stop codon must immediately precede the extension in transcript coordinates.

### Readthrough rates

Stop codon readthrough rates were evaluated by dividing the ribosome density (in RPKM) for each C-terminal extension by the ribosome density in the corresponding CDS. In cases where multiple transcript isoforms contained the same extension, the transcript that minimized the ratio of ribosome footprint density in the extension to the density in the CDS was reported. To control for variability introduced by start and stop codon peaks (see *Figure 2—figure supplement 1*), we excluded the following genomic positions from our totals: 12 nucleotides following the start codon, the 15 nucleotides preceding the stop codons of the coding region and the extension, and the 9 nucleotides following the stop codon of the coding region.

### Scoring of predicted extensions

A predicted extension was scored positively if: (a) there existed ribosome density in the extension, (b) ribosome density vanished or unambiguously decreased after the extension's in-frame stop codon, and (c) positions occupied by ribosomes in the readthrough region were evenly-spaced throughout the extension. When ribosome density was sparse in the extension, we relaxed criterion (c) and additionally required a peak of at least two reads at the extension's stop codon. Aside from *Kelch,* which has been demonstrated to undergo readthrough experimentally (*Robinson and Cooley, 1997*), we did not positively score any extension that contained a methionine in its first three codons, as these could represent downstream cistrons rather than true extensions. Furthermore, we required read density upstream of the first methionine in any extension containing a methionine codon.

## Identification of novel extensions

We identified all coding transcripts in FlyBase r5.43 in which: (a) the 3′ UTR was annotated, (b) a C-terminal extension was not predicted by Jungreis et al, (c) there were at least five codons between the annotated stop codon of the CDS and the next in-frame stop codon, and (d) the region between these stop codons (the putative extension) did not overlap any annotated CDS, 5′UTR, tRNA, rRNA, snRNA, snoRNA, miRNA, or pre-miRNA. We additionally excluded extensions whose translation could be explained by alternative splice isoforms whose transcripts omitted the stop codon, using splice junctions from FlyBase revision 5.43 and inferred from our on RNA-seq data, as described above.

Following the same scoring criteria we used for the extensions predicted by Jungreis et al., we scored each candidate extension that met the following criteria: (a) a minimum read density of 0.2 RPKM in the extension, (b) a minimum readthrough rate of 0.001, (c) at least 10% of the nucleotide positions in the extension covered by reads, (d) the first read occurring within the first quartile of extension length, (e) the last read occurring within last quartile of the extension length, and (f) a 75% or greater decrease in read density in the first 114 nucleotides of distal 3′ UTR compared to the extension. To calculate this last statistic for transcripts whose distal 3′ UTR was less than 114 nt in length, we extended the distal 3′ UTR in uninterrupted genomic coordinates to 114 nt in length.

## Metagene analyses

For each analyses (*Figure 2—figure supplement 1*, *Figure 4B*), we identified regions of interest (ROIs) germane to the analysis. In *Figure 2—figure supplement 1*, these included roughly 3000 ROIs each for the left and right panels, each of which met the following criteria: (a) all transcripts deriving from that gene had one annotated start codon (left panel) or stop codon (right panel), (b) all transcripts deriving from that locus covered identical genomic positions over the region of interest (ROI) shown, (c) all positions within the ROI were non-degenerate (see 'Materials and methods'), and (d) at least 10 reads were present in the coding subregion of the ROI. For coding regions in *Figure 4C*, we kept the same criteria as above but required only 0.5 reads in the coding subregion of each ROI, yielding roughly 7401 ROI for that set. For C-terminal extensions, we required only that the extension be long enough to cover the interval shown, and have 0.5 reads in the coding subregion, allowing us to include 123 of the 350 extensions.

For each ROI, we then generated a 'coverage vector' tallying ribosome density at each nucleotide position. We then normalized each coverage vector to the mean number of footprint reads covering the annotated coding region in the ROI, excluding a 3-codon buffer flanking the start or stop codon to avoid bleedthrough from initiation or termination peaks. We then plotted the median value across all normalized coverage vectors at each position.

## Search for genomic polymorphisms and A-to-I editing

To improve our sensitivity in detection, we re-aligned our footprint and mRNA datasets to a Bowtie database of spliced transcript models, allowing three mismatches (where we previously only allowed two). For the first, second, and third nucleotide position in each unique, annotated stop codon, we counted the number of matching and mismatching nucleotides in each read alignment covering that position. We ignored mismatches that occurred in the first position of the read alignment, because they frequently arise from non-templated nucleotide addition by reverse transcriptase. Considering the first, second, and third positions of each stop codon separately, we calculated a global average mismatch frequency for each. We then searched for individual stop codon positions that far exceeded the corresponding global average using a binomial test, controlling the false discovery rate at 5% following the procedure of Benjamini and Hochberg (*Benjamini and Hochberg, 1995*). We performed this analysis separately upon each of three datasets: total mRNA, total footprints, and the subset of footprints whose P-sites had passed the nucleotide position in question, following the P-site assignment rules described above.

## Immunoprecipitation and western blotting

4–48 ml of transiently or stably transfected cells were harvested 48 or 72 hr post-transfection by centrifuging for 2 min at 1600 rpm in a tabletop centrifuge. All cell pellets were rinsed once in PBS, and flash-frozen in a bath of dry ice in ethanol. Cell pellets were thawed and lysed for at least 15 min on ice in 0.5–1.5 ml lysis buffer (150 mM NaCl, 50 mM Tris pH 7.5, 1% Triton x-100, 1 mM EDTA and 1× complete protease inhibitor cocktail [Roche Applied Science]), depending upon the pellet size. Lysates were clarified by spinning 10 min at 20,000 × *g* in a microcentrifuge and supernatants

collected. GFP reporters were immunoprecipitated on 10 μl of anti-GFP beads (Chromotek, Planegg-Martinsried, Germany) equilibrated in IP wash buffer (150 mM NaCl, 50 mM Tris pH 7.5, 1 mM EDTA, 0.05% Triton x-100). The bound fraction was washed three times in IP wash buffer, and finally eluted by boiling for at least 5 min in NuPage sample loading buffer (Invitrogen). Supernatants were collected and transferred to new tubes, and stored at −20°C.

For western blotting, samples were resolved on 4–12% NuPage gels (Invitrogen) in MOPS buffer. Gel lanes were loaded such that the amounts of uncleaved GFP reporter in each lane were loaded as equally as possible. GFP was detected using a mouse anti-GFP antibody (Roche Applied Sciences), and visualized using IR800 anti-mouse antibodies on a LI-COR Odyssey system (LI-COR, Lincoln, Nebraska). FLAG was similarly detected on a separate gel, instead using the M2 Mouse anti-FLAG antibody (Sigma-Aldrich).

## PhyloCSF analysis

PhyloCSF analysis was performed on all C-terminal extensions at least five codons long, exclusive of the stop codons. Multiple species alignments were obtained from the *Drosophila* 12-way multispecies alignment as downloaded from the UCSC genome browser, and stitched together over regions of interest using the Phast utility maf_parse (*Hubisz et al., 2011*). PhyloCSF was then used to evaluate the extension on the empirical codon model '12flies'. Columns in which the *D. melanogaster* sequence contained gaps were ignored. Alignments that contained no sequence besides that from *D. melanogaster* were not evaluated.

## Z-Curve classifier

We calculated the 189-variable Z-curve as previously described (*Gao and Zhang, 2004*). We empirically determined that the classifier became error-prone if trained on sequences 81 nt or shorter in length. Our training set consisted of 81-nucleotide windows drawn from coding regions (the positive set, 14,507 windows) or from portions of distal 3′ UTRs that did not overlap annotated coding regions or 5′ UTRs (the negative set, 8151 windows). To assay the stability of the classifier's behavior and control for overfitting, we trained the classifier with fourfold cross-validation training on 2200 windows from the CDS set and 2200 windows from the distal 3′ UTR set, yielding an average misclassification error of 6.9–7.3% with each iteration. We repeated this analysis (and cross-validation) several times selecting different 81-nucleotide windows from each CDS and distal 3′ UTR, obtaining similar levels of error. The classifier was then trained on the entire training set, and used to evaluate randomly chosen 81 nt windows from observed C-terminal protein extensions that were 81 nt or greater in length. These included 26 extensions predicted by Jungreis et al. and 83 novel extensions.

## SNP analysis

We downloaded SNP data from the *Drosophila* Population Genomics Project from Ensembl.org (release 67; *Flicek et al., 2013*) and counted the proportion of SNPs that, if translated in frame, would cause synonymous substitutions in coding regions, extensions, and distal 3′ UTRs.

## Tests for differential regulation of readthrough rates

To test C-terminal extensions for differential readthrough rates, we examined all extensions which met the following criteria: (1) all annotated isoforms covering the extension contain exactly the same CDS, (2) the CDS had at least 128 total footprint reads in each of the S2 cell and embryo samples, and (3) the C-terminal extension had been scored as positive for readthrough in either the S2 and/or the embryo sample. For those extensions, we tabulated the footprint reads that aligned to the CDS and putative extension, masking out regions normally covered by start and stop codon peaks as described (see section 'Readthrough rates', above). For each extension, this tabulation yielded a 2 × 2 contingency table of reads aligning to the CDS and extension in the S2 cell and 0–2 hr embryo datasets. We evaluated the statistical significance of asymmetry in the contingency tables using Fisher's exact test, and controlled the false discovery rate at 5% using the procedure of Benjamini and Hochberg (*Benjamini and Hochberg, 1995*).

## Human and yeast ribosome profiling data

For human cells, we collected data from uninfected human foreskin fibroblasts and processed it as previously described (*Stern-Ginossar et al., 2012*). Yeast samples were collected from [*psi*−] W303 and processed as previously described (*Ingolia et al., 2009*), with the exception that a 3′ linker ligation

strategy was used instead of poly(A) tailing for fragment capture. For phasing of yeast footprints, we counted only 28-mers, which have previously been shown to be the best-phased footprint population in that organism (*Ingolia et al., 2009*).

## Motif prediction

C-terminal extensions 20 amino acids or longer were scanned for transmembrane domains using TmHmm (*Krogh et al., 2001*) using default settings. Nuclear localization signals were predicted in extensions 20 amino acids or longer using the cNLS mapper (*Kosugi et al., 2009*), with a score cutoff of 7.0. Peroxisome targeting signals were predicted for all extensions 12 amino acids or longer using PTS1 Predictor (*Neuberger et al., 2003*) with the signal type set to 'metazoan'. Prenylation signals were predicted for all extensions 12 amino acids or longer using PrePS (*Maurer-Stroh and Eisenhaber, 2005*). In addition, we searched for ER retention signals using the consensus [KH]DEL*.

We searched 3′ UTRs (including the predicted extension and entire distal 3′ UTR) for selenocysteine insertion elements using SeciSearch 2.19 (*Kryukov et al., 2003*) with parameters set as follows: e1 = 05, e2 = −22, Y_filter = True, O_filter = True, B_filter = True, S_filter = True. We searched each 3′ UTR using every available SECIS Pattern (pat_c, pat_Sep20, pat_dm, pat_g, and pat_s), and considered a 3′ UTR receiving a COVE score above the recommended threshold of 15 in any of the pattern searches to contain a SECIS element. Additionally, we excluded any extensions that were annotated as selenoprotein annotations in SelenoDB (*Castellano et al., 2008*; for *Drosophila,* yeast, and human data) or FlyBase (*Marygold et al., 2013*; for *Drosophila*), For transcripts with no annotated or short 3′ UTRs, we extended the 3′ UTR in uninterrupted genome coordinates until it was 1000 nucleotides in length, an in *Jungreis et al. (2011)*.

## Microscopy

S2 cells stably transfected with the reporter of interest were maintained at a density of 1.6–12 million cells/ml. Nuclei were visualized by staining with 1 µg/ml Hoechst 34580 (Invitrogen) for at least 5 min. Live cells were imaged in culture media on an inverted spinning disk confocal Nikon Ti microscope (Nikon Instruments, Melville, NY) in glass-bottom culture dishes (MatTek, Ashland, MA). Images were contrast-adjusted and prepared for presentation in Adobe Photoshop (Adobe Systems, San Jose, CA).

## Data files

Both the raw data (as FastQ files) and processed data (wiggle files) are available in NCBI's Gene Expression Omnibus (*Edgar et al., 2002*) under GEO series accession number GSE49197 (http://www.ncbi.nlm.nih.gov/geo/query/acc.cgi?acc=GSE49197). Supplementary tables 1–4 are available at Dryad (*Dunn et al., 2013*; http://dx.doi.org/10.5061/dryad.6nr73):

Supplementary table 1: gene expression measurements in 0–2 hr embryos and S2 cells. Source data for *Figures 1 and 2*, as well as their supplements.

Supplementary table 2: readthrough statistics for *Drosophila melanogaster*. Source data for *Figures 3, 4 and 6*, as well as their supplements, and annotations of readthrough events in *Drosophila melanogaster*.

Supplementary table 3: readthrough statistics for *Saccharomyces cerevisiae*. Source data for *Figure 5* and annotations of readthrough events in [*psi*] W303 yeast.

Supplementary table 4: readthrough statistics for human foreskin fibroblasts. Source data for *Figure 5* and annotations of readthrough events in human foreskin fibroblasts.

*Supplementary file 1* provides alignment statistics and *Supplementary file 2* contains the oligonucleotides used in this study.

## Other software and libraries

We wrote custom scripts in *Python* 2.7, using the following open-source libraries: Numpy 1.6.0 (http://numpy.scipy.org), Scipy 0.11.0rc2 (http://www.scipy.org), Biopython 1.59 (*Cock et al., 2009*), PySam, and HTSeq 0.5.1p2 (http://www-huber.embl.de/users/anders/HTSeq/doc/overview.html). Plots and genome browser snapshots were generated using Matplotlib 1.0.1 (*Hunter, 2007*).

## Acknowledgements

We thank John Atkins for pointing out the importance of translation readthrough in *Drosophila;* Irwin Jungreis and Manolis Kellis for critical comments on the manuscript; Noam Ginossar for supplying the

human foreskin fibroblast ribosome profiling data; Jeffrey Farrell, Tony Shermoen, and Hansong Ma for useful conversation and help with handling flies; Onn Brandman, Luke Gilbert, Noam Ginossar, Calvin Jan, Gene-wei Li, and Eugene Oh for advice on laboratory and analytical methods; and Clement Chu, Jessica Lund, and Silvi Rouskin for help with sequencing. We also thank Jessica Walter, the UCSF Cell Propulsion Lab, DeLaine Larsen, and the Nikon Imaging Center at UCSF for help with imaging.

## Additional information

### Funding

| Funder | Grant reference number | Author |
| --- | --- | --- |
| Howard Hughes Medical Institute | | Joshua G Dunn, Catherine K Foo, Jonathan S Weissman |
| National Science Foundation | Graduate Research Fellowship | Joshua G Dunn |
| National Institutes of Health | GM061107 | Nicolette G Belletier, Elizabeth R Gavis |
| National Institutes of Health | P50 GM102706 | Joshua G Dunn, Catherine K Foo, Jonathan S Weissman |

The funders had no role in study design, data collection and interpretation, or the decision to submit the work for publication.

### Author contributions

JGD, Performed *Drosophila* ribosome profiling, Developed bioinformatic methods, Conception and design, Acquisition of data, Analysis and interpretation of data, Drafting or revising the article, Contributed unpublished essential data or reagents; CKF, Performed yeast ribosome profiling, Contributed unpublished essential data or reagents; NGB, ERG, Drafting or revising the article, Contributed unpublished essential data or reagents; JSW, Conception and design, Analysis and interpretation of data, Drafting or revising the article

## Additional files

### Supplementary files

• Supplementary file 1. Alignment statistics. Provides statistics on read alignments by sample and genomic region (e.g., CDS, 5′ UTR, 3′ UTR, intergenic, etc; **A**), as well as by sample and alignment type (e.g., chromosomal, spliced, unaligned; **B**).
• Supplementary file 2. Oligonucleotides used in this study. For readers who wish to implement the *Drosophila* ribosome profiling protocol.

### Major datasets

The following datasets were generated:

| Author(s) | Year | Dataset title | Dataset ID and/or URL | Database, license, and accessibility information |
| --- | --- | --- | --- | --- |
| Dunn JG, Weissman JS | 2013 | Ribosome profiling reveals pervasive and regulated stop codon readthrough in *Drosophila melanogaster* | http://www.ncbi.nlm.nih.gov/geo/query/acc.cgi?acc=GSE49197 | Publicly available at NCBI GEO (http://www.ncbi.nlm.nih.gov/geo/). |
| Dunn JG, Foo CK, Belletier NG, Gavis ER, Weissman JS | 2013 | Data from: Ribosome profiling reveals pervasive and regulated stop codon readthrough in *Drosophila melanogaster* | http://dx.doi.org/10.5061/dryad.6nr73 | Supplementary tables 1–4 publicly available at Dryad (http://www.datadryad.org). |

The following previously published datasets were used:

| Author(s) | Year | Dataset title | Dataset ID and/or URL | Database, license, and accessibility information |
|---|---|---|---|---|
| Drosophila Population Genomics Project | Downloaded from Ensembl on 23 August, 2012 | Data from: 50 Genomes - release 1.0 | ftp://ftp.ensembl.org/pub/release-67/variation/gvf/drosophila_melanogaster/Drosophila_melanogaster.gvf.gz | Freely available online |
| The Flybase Consortium | | Data from: *D. melanogaster* genome annotation revision 5.43. | ftp://ftp.flybase.net/genomes/Drosophila_melanogaster/dmel_r5.43_FB2012_01/gff/dmel-all-no-analysis-r5.43.gff.gz | Freely available online |
| Jungreis I, Lin MF, Spokony R, Chan CS, Negre N, Victorsen A, White KP, Kellis M | 2011 | Data from: Evidence of abundant stop codon readthrough in *Drosophila* and other metazoa (*Genome Res.* 2011 21: 2096-2113) | http://genome.cshlp.org/content/suppl/2011/09/28/gr.119974.110.DC1/Data1_DmelReadthroughCandidates.txt | Freely available online through the *Genome Research* Open Access option (Supp Data1.txt). |
| Qin X, Ahn S, Speed TP, Rubin GM | 2007 | Data from: Global analyses of mRNA translational control during early *Drosophila* embryogenesis (*Genome Biology* doi:10.1186/gb-2007-8-4-r63) | http://genomebiology.com/content/supplementary/gb-2007-8-4-r63-s1.xls | Open data from *Genome Biology* (Additional data file 1). |
| Saccharomyces Genome Database project | Downloaded 22 January 2013 | Data from: Genome Release R64-1-1 and corresponding gene annotation | http://downloads.yeastgenome.org/curation/chromosomal_feature/saccharomyces_cerevisiae.gff | Freely available online |

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
