## [Decision Letter]

Thank you for sending your work entitled “Ribosome profiling reveals pervasive and regulated stop codon readthrough in *Drosophila melanogaster*” for consideration at *eLife*. Your article has been favorably evaluated by a Senior editor and 3 reviewers, one of whom, Nahum Sonenberg, is a member of our Board of Reviewing Editors.

The consensus opinion of the reviewers is that this study is a thorough, compelling analysis that describes the development of a ribosome profiling assay for *Drosophila melanogaster* and provides the first genome-wide experimental analysis of stop codon readthrough. The data support most of your conclusions.

The important conclusions are that:

A) Readthrough is more pervasive than expected, and the majority of readthrough events observed were not predicted phylogenetically.

B) The C-terminal protein extensions show evidence of selection, contain functional subcellular localization signals, and their readthrough is regulated, arguing for their importance.

C) The readthrough might regulate gene expression and protein function, and to add plasticity to the proteome during evolution.

However, the reviewers raised several concerns and questions as described below.

1) You note that the locations of ribosome-protected footprint fragments from yeast and human ribosome profiling datasets exhibit 3-nucleotide periodicity from which reading frames can be deduced. Does the fly data provide enough resolution to also show such periodicity? If not, why?

2) The reviewers agree that you need to indicate more clearly the mean level of readthrough you observe with the predicted and novel extensions. You could have discussed this in a couple of places, but you didn't seize upon those opportunities. For example, Figure 5 appears to show that the mean readthrough rates observed range from 1-3%. However, only a couple of rather cryptic references that addressed this point are in the text. In the Results section you state that the human, yeast and fly samples cover similar ranges of efficiency. In the Discussion, you state that the readthrough rates range beyond 10%, while the baseline readthrough is much lower (0.0-1.4%). You need to address the mean level of readthrough more directly in the text since the level of readthrough you observe directly relates to your functional significance arguments in the Discussion. Have you tried to show that one of the newly discovered endogenous proteins has a longer isoform than the predicted size encoded by its corresponding coding region? If not, this needs to be noted.

3) Recent studies have shown that dedicated recycling factors (Rli1 in yeast and ABCE1 in mammals) are required for efficient ribosome release following translation termination. A key concern related to your predicted and novel extensions is whether the ribosomes distal to the stop codon represent translating ribosomes, or simply ribosomes that may not have properly released from the mRNA following termination at the stop codon. In a recent Cell paper (23), a parameter called the Ribosome Release Score (RRS) was used to discriminate between translated protein-coding regions and non-coding transcripts with similar ribosome densities. Could you apply that parameter to the stop codons of the predicted and novel extensions to provide further confidence that you truly represent translated extensions, rather than 3´-UTRs that simply don't release ribosomes? The Guttman paper should be cited.

4) Discussion: You state that readthrough is pervasive, biologically regulated, and functionally consequential, and thus provides an important mechanism to regulate gene expression and function. In light of my concern about the level of readthrough you are generally observing (1-3%), the reviewers think this is somewhat overstating your results. Until you have eliminated one or more of these C-terminal extensions and shown that it results in an adverse phenotype, you cannot say that these extensions are “functionally consequential”.

5) The evolutionary analysis presented near the end of the paper is problematic.

You partitioned the observed examples of readthrough into those that had been predicted by Lin et al. to have signatures of coding conservation (predicted readthrough) and those that didn't (novel readthrough). Then you use three pieces of data to argue that the novel readthroughs are under purifying selection to maintain their coding capacity, and that they are of recent evolutionary origin.

First, you used PhyloCSF to score the novel readthrough, finding that few score positively and that you have the same distribution of scores as non read-through 3'UTRs. You posit that there are only two possible explanations for this - that the novel readthrough are selectively neutral, or that you are of too recent origin to leave a detectable phylogenetic signature. But it is also possible that you have a signature, but it is simply too weak to detect with the model used by PhyloCSF. We come back to this point below.

Next, you compared predicted and novel readthrough, UTRs from non read-through genes, and coding sequence using an algorithm that separates coding and non-coding sequence using nucleotide frequencies, finding that novel readthrough were somewhere in between coding and predicted readthrough on the one hand and non-coding sequence on the other. You note that this is consistent with an “evolutionary trajectory” from non-coding to coding. However, it is also consistent with sequences that simply have weak coding propensity.

Finally, you look at *D. melanogaster* SNPs to evaluate whether there is a preference for synonymous relative to non-synonymous SNPs in novel vs predicted read-through, finding that there is a weak preference for synonymous SNPs, less than found in predicted SNPs, and say this is consistent with “mild or recently-imposed selection”. This might not be correct. If you have two read-through events – one which evolved at the base of the genus *Drosophila*, and one along the lineage that separates *D. melanogaster* from *D. simulans*/*D. sechellia* – and posit that these read-through events are under identical selective pressure, then you would expect both to have identical preference for synonymous substitutions, regardless of when you evolved. There is good reason to assume that recently evolved sequences would be under any different strength of selection. If this novel hypothesis were correct, you would probably expect some, if not most, of the novel events to be polymorphic within the population, with some of these in the middle of selective sweeps. But, this would produce a different synonymous vs non-synonymous pattern (with an excess of non-synonymous SNPs perhaps).

It is odd to argue that the majority of read-through events are novel and under selection – something that you'd only expect to find if readthrough events had very short evolutionary half-lives or if there were some reason to have specifically evolved functional read-through events in *D. melanogaster*.

The data are equally, if not more, consistent with the novel readthrough being subject to weak selection, with their origins unknown. One simple way to resolve this might be to run the Z-score program on orthologous UTRs of read-through and non-read-through genes across the genus. If these are novel to *D. melanogaster*, their scores should be significantly higher in *D. melanogaster*.

---

## [Author Response]

*1) You note that the locations of ribosome-protected footprint fragments from yeast and human ribosome profiling datasets exhibit 3-nucleotide periodicity from which reading frames can be deduced. Does the fly data provide enough resolution to also show such periodicity? If not, why*?

The fly data do not provide sufficient resolution to show periodicity. Because our standard nuclease, RNase I, destroys *Drosophila* ribosomes, we prepared the fly libraries with micrococcal nuclease (MNase), which *Drosophila* ribosomes tolerate well over a wide range of concentrations (see Figure 1—figure supplement 1). While RNase I is an unbiased enzyme, MNase has a strong 3' A/T bias. As a result, MNase-digested footprints are longer than RNase-digested footprints, and not always fully resolved to the edges of ribosomes. This fact gives rise to a small amount of positional uncertainty with P-site mapping in MNase datasets. We handle this uncertainty in our P-site assignment strategy by assigning a fraction of the P-site over a neighborhood of adjacent nucleotides determined by the length and endpoints of each read alignment as detailed in the methods section of our manuscript and as previously performed in [60]. We had already included a discussion of mapping in the Materials and methods (section “Sequence processing and alignment”), but have now included an explicit discussion of why periodicity is not visible on fly data in two places in our main text.

*2) The reviewers agree that you need to indicate more clearly the mean level of readthrough you observe with the predicted and novel extensions. You could have discussed this in a couple of places, but you didn't seize upon those opportunities. For example,*
Figure 5
*appears to show that the mean readthrough rates observed range from 1-3%. However, only a couple of rather cryptic references that addressed this point are in the text. In the Results section you state that the human, yeast and fly samples cover similar ranges of efficiency. In the Discussion, you state that the readthrough rates range beyond 10%, while the baseline readthrough is much lower (0.0-1.4%). You need to address the mean level of readthrough more directly in the text since the level of readthrough you observe directly relates to your functional significance arguments in the Discussion*.

We have added specific references to median levels of readthrough observed in the text. To set a threshold for biological significance, we now discuss readthrough rates in comparison to the distribution we observe for the phylogenetically conserved extensions in the *Drosophila* embryos. The rationale is as follows: because the phylogenetically conserved readthrough events are likely to be conserved because they are functional, and because only a specific fraction of these extensions have observable amounts of readthrough in our samples, we infer that this specific group, when translated, is translated at a rate that is biologically functional. We therefore re-framed our text to explicitly compare readthrough rates of various extensions against this phylogenetically conserved group of extensions in *Drosophila melanogaster.* We note this first when discussing readthrough in yeast and humans:

“To estimate how many of the novel extensions we detected might be translated at a biologically significant level…”

And we explicitly state the median readthrough rates we observe in our Discussion:

“Finally, our model predicts that conserved extensions should on average exhibit higher readthrough rates than novel extensions…”

*Have you tried to show that one of the newly discovered endogenous proteins has a longer isoform than the predicted size encoded by its corresponding coding region? If not, this needs to be noted*.

We have amended the text to indicate that we did not seek to detect endogenous proteins. The readthrough reporters we designed for Figure 4 contain 120 codons upstream of the annotated stop codon, and the entire endogenous 3' UTR, the latter modified only to include a double FLAG epitope upstream of the extension's termination codon. Because we included so large a region of the endogenous mRNA in our constructs, we believe that they report readthrough at least as faithfully as those traditionally used in the literature to screen readthrough contexts, which include much less endogenous sequence (as few as 2–8 codons upstream and 3-15 codons downstream of the stop; Fearon et al*.*, 1994; Feng et al*.*, 1992; Harrell et al*.*, 2002; Namy et al*.*, 2002; Namy et al*.*, 2003), with the exception of a number of excellent papers that deduce the minimal requirements for readthrough in specific virus or host transcripts (Cimino et al*.*, 2011; Firth et al*.*, 2011; Napthine et al*.*, 2012; Skuzeski et al*.*, 1991; [69]).

*3) Recent studies have shown that dedicated recycling factors (Rli1 in yeast and ABCE1 in mammals) are required for efficient ribosome release following translation termination. A key concern related to your predicted and novel extensions is whether the ribosomes distal to the stop codon represent translating ribosomes, or simply ribosomes that may not have properly released from the mRNA following termination at the stop codon. In a recent Cell paper (*[23]*), a parameter called the Ribosome Release Score (RRS) was used to discriminate between translated protein-coding regions and non-coding transcripts with similar ribosome densities. Could you apply that parameter to the stop codons of the predicted and novel extensions to provide further confidence that you truly represent translated extensions, rather than 3´-UTRs that simply don't release ribosomes? The Guttman paper should be cited*.

We agree that this is an important point. However, it is important to note that RRS score as developed and validated in [23] is not applicable for this specific purpose. The authors state explicitly that RRS is best suited to classify a transcript as containing or lacking a single, predominant long open reading frame, not to choose which stop codon in that transcript is utilized:

“RRS is not designed to identify specific translated regions within a transcript containing multiple overlapping or nearby translated regions” (Guttman et al*.*, 2013).

Therefore, in our manuscript we sought to control for this possibility in two other ways.

First, we required a 75% or greater decrease in ribosome density following the first in-frame stop codon as a preliminary filtering criterion before a given C-terminal extension was even examined for readthrough (see Materials and methods section “Identification of C-terminal protein extensions” subsection “Identification of novel extensions”).

Second, we demonstrated in a metagene analysis that ribosome footprint density covering the stop codons that terminate C-terminal extensions extensions is qualitatively similar to the density covering stop codons of annotated coding regions (Figure 4). In this analysis, clear termination peaks are visible over stop codons in both metagene averages, and, in each average, the normalized ribosome footprint density drops to negligible levels after the stop codon in question. Thus, the fact that ribosomes occupying the extensions show characteristic behaviors of termination (spikes at the stop codon, followed by a drop in density) at the C-termini of the extensions strongly argues that those ribosomes are engaged in active translation of extensions, rather than just sliding.

Nonetheless, we have included in this revised manuscript an additional analysis of ribosome release similar to the RRS score (Figure 4—figure supplement 1). Briefly, we tabulate the ratio of reads in a 5-codon window downstream of a given stop codon to the number of reads in a 5-codon window upstream of that stop codon. This criterion differs from the various ways RRS was calculated in Guttman et al. principally in that RRS is additionally normalized for mRNA fragment density to control for transcript mis-annotation (Guttman et al*.*, 2013), something we did not consider in our study as we manually verified the structures of all transcripts for which we report readthrough.

We perform this RRS-like calculation on the following classes of codons: 1) stop codons that terminate annotated coding regions, 2) stop codons that terminate C-terminal extensions, and 3) randomly-selected codons internal to annotated coding regions. We find that the release scores for C-terminal extensions fall well within the distribution for those of annotated coding regions, which again supports the notion that the ribosome footprint density covering extensions represents bona fide translation events, followed by termination at the expected stop codon. In addition, we have cited both [23] and [66] and expanded our discussion of ribosome release to improve clarity of this issue:

*4) Discussion: You state that readthrough is pervasive, biologically regulated, and functionally consequential, and thus provides an important mechanism to regulate gene expression and function. In light of my concern about the level of readthrough you are generally observing (1-3%), the reviewers think this is somewhat overstating your results. Until you have eliminated one or more of these C-terminal extensions and shown that it results in an adverse phenotype, you cannot say that these extensions are “functionally consequential”*.

We appreciate this objection and have changed our language accordingly. The sentence now reads:

“Our studies indicate that readthrough is far more pervasive than previously appreciated, is biologically regulated, and may append functional peptide signals to host proteins.”

Nonetheless, it is worth noting that finding a phenotype for a given protein is difficult. The *Drosophila* gene *kelch,* the first gene discovered to undergo readthrough in *Drosophila,* provides a good example. *kelch* is essential for female fertility, and has an extension notable for both its conservation (PhyloCSF score 7784, conserved throughout the sequenced *Drosophila* phylogeny) and length (787 amino acids). However, Robinson and colleagues found that expression of specifically the short form, but not of the long form, complemented the fertility defect observed in the null mutant (65). Nonetheless, given its conservation and length, it is hard to imagine that this extension is not functional. Robinson and colleagues therefore proposed that the long form of *kelch* may be more important in other tissues (e.g., the imaginal discs, where the long form is specifically up-regulated relative to the short form), but they did not create the conditional mutants necessary to test this hypothesis (65).

That said, we are very interested in investigating the functions of specific extensions, and will probably start this work in yeast, which offers sophisticated genetic tools, a simpler life history, and several C-terminal extensions in essential genes that might yield interesting phenotypes. We hope this will provide fertile ground for future studies.

*5) The evolutionary analysis presented near the end of the paper is problematic*.

We recognize these concerns and have adjusted our language in the text to highlight alternate interpretations of the data per the reviewers' concerns (discussed further below). In this revised manuscript we try to make clear the evolutionary model we present is a more speculative part of the discussion. As with any evolutionary question, multiple explanations may account for our observations. We present in the revised text what we believe to be a reasonable and plausible explanation. Importantly, we do not wish to overstate our results or to give false impressions of certainty. Rather, we hope our work will provide a point of entry into further investigations on the origins and functions of C-terminal extensions.

*You partitioned the observed examples of readthrough into those that had been predicted by Lin et al. to have signatures of coding conservation (predicted readthrough) and those that didn't (novel readthrough). Then you use three pieces of data to argue that the novel readthroughs are under purifying selection to maintain their coding capacity, and that they are of recent evolutionary origin*.

*First, you used PhyloCSF to score the novel readthrough, finding that few score positively and that you have the same distribution of scores as non read-through 3'UTRs. You posit that there are only two possible explanations for this – that the novel readthrough are selectively neutral, or that you are of too recent origin to leave a detectable phylogenetic signature. But it is also possible that you have a signature, but it is simply too weak to detect with the model used by PhyloCSF. We come back to this point below*.

We regret that our manuscript was unclear on an important point: our goal was to evaluate — regardless of evolutionary age — whether any of the novel readthrough events we identified occur because they are biologically important or, alternatively, simply because they can occur without incurring a significant fitness disadvantage (i.e., are selectively neutral or nearly neutral). The first way we approached this question was to look for evidence of selection, the presence or absence of which would favor one hypothesis over the other.

We did interpret the novel extensions we identified to be, on average, evolutionarily recent in origin because of their negative PhyloCSF scores. In so doing, we made two assumptions. First, we assumed that PhyloCSF's model for the *Drosophila* phylogeny should be sensitive to detect conservation among most, even if not all, conserved coding regions. Secondly, we assumed that protein-coding selection, if present, should yield similar signatures in the amino acid sequences of known coding regions and of putative C-terminal extensions (i.e., conservation, if present, should favor synonymous amino acid substitutions over non-synonymous changes, and the primary amino acid sequence should be important).

This first assumption is consistent with performance benchmarks of PhyloCSF, which demonstrate that it detects signatures of protein coding conservation 93% of known *Drosophila* coding regions in *Drosophila* (Lin et al*.*, 2011). Even when restricted to short (10–60 codon) regions, which are notably difficult to evaluate (Lin et al., 2008), PhyloCSF still achieves greater than 90% sensitivity, with roughly 98% specificity (Lin et al*.*, 2011). Given that the novel extensions we report largely fall within this size range (median: 16 codons; 76% 10 codons or longer), we think that a signal of conservation, if present, should be detectable by PhyloCSF with comparable sensitivity (provided our second assumption, that extensions should be have similarly in their conservation properties to known coding regions, is also reasonable; further discussed below). Therefore, while some conserved signals may score negatively purely by chance, the majority of conserved signals should score positively.

In the specific case that selection is present but sufficiently weak to be undetectable, we would expect PhyloCSF scores to approach zero. This expectation arises from the fact that the PhyloCSF score is actually a likelihood ratio, calculated as the log ratio of probabilities of observing a given set of triplet substitutions under a coding model of evolution versus a non-coding model of evolution. Therefore, positive scores indicate a higher probability of observing a given set of substitutions if the ancestral sequence were coding, while a negative score actually indicates a greater likelihood of observing those substitutions under a non-coding model. In other words, a negative score actually provides evidence in favor of a non-coding model rather than indicate the absence of evidence for a protein-coding model. Simultaneous absence of evidence of both models would a ratio of probabilities close to 1.0 and a log-ratio of 0. Instead, we find a median PhyloCSF score for the novel extensions to be -159.8 decibans, indicating that, on average, a non-coding model fits these extensions far better (10^16^-fold) than a coding model. Despite this fact, in our manuscript, we deliberately use more conservative language and merely note a “lack of phylogenetic evidence for amino acid conservation,” rather than “evidence against conservation of protein coding.”

Our second assumption is that phylogenetic conservation should exhibit the same signals for C-terminal protein extensions as for classical protein-coding regions. We think this assumption to be reasonable because a large group of extensions — 283 proposed by Jungreis et al., 43 confirmed by us — conform to this behavior, and because, to our knowledge, the circumstances in which conservation does not yield such phylogenetic signatures are few.

However, such circumstances do exist. The prion-forming domains of fungi, in which prion forming ability can be maintained even if primary amino acid sequence is scrambled, provide an example (Ross et al*.*, 2005). It is possible that some of the extensions we have identified fall into a similar category, but we believe this number not to be large, as there are few known protein domains that behave similarly to the fungal prion-forming domains in this particular regard.

Another circumstance in which an ancient extension might score negatively by PhyloCSF is one in which the act of reading through a stop codon is ancient and phylogenetically conserved, but the amino acid sequence of the resulting extension unimportant, relatively unconstrained, and evolving. This scenario is similar to, but less constrained than, the case of the prion-forming domains in fungi, but supposes that either the act of readthrough or signals that incidentally promote readthrough, rather than readthrough products, are what has been selected. This circumstance provides a specific explanation for why an extension might be unselected (and appear to be selectively neutral or nearly-neutral, a model we already account for), rather than another model to explain the data.

Notably, in this scenario, if an evolutionarily unconstrained extension acquires a biological function that is subsequently selected and fixed, the extension and its function may be reasonably interpreted to be evolutionarily novel even if the act of readthrough at the upstream stop codon is ancient, because the extension's primary amino acid sequence and the function it yields are in fact evolutionarily novel. This circumstance is in fact a special case of the evolutionary model we propose in our Discussion, in which there is simply a long time lag between selection upon the extension and the appearance of a readthrough event.

*Next, you compared predicted and novel readthrough, UTRs from non read-through genes, and coding sequence using an algorithm that separates coding and non-coding sequence using nucleotide frequencies, finding that novel readthrough were somewhere in between coding and predicted readthrough on the one hand and non-coding sequence on the other. You note that this is consistent with an “evolutionary trajectory” from non-coding to coding. However, it is also consistent with sequences that simply have weak coding propensity*.

These interpretations are mutually consistent. Under the Z-curve model, a weak coding propensity corresponds to a Z-curve score that is intermediate between the CDS-like and 3'UTR-like distributions. Our goal is to evaluate explanations for why these sequences, as a group, would show such a distribution of weak coding propensities. One explanation is the evolutionary trajectory we describe in our manuscript. Another plausible explanation would be the inverse trajectory, from a CDS-like nucleotide character to a 3'UTR-like nucleotide character. This situation would be expected to occur if the novel extensions we identified by ribosome profiling were caused by recent acquisition of a stop codon somewhere in the *melanogaster* lineage, followed by degradation of the formerly-coding sequence downstream of that stop. However, in this case, these extensions would be expected to: a) on average score positively by PhyloCSF (because they would be conserved in other, more ancient lineages), and b) lack upstream stop codons in species other than *melanogaster.* They do not. We therefore do not favor this model.

A third model, that the novel extensions as a group show a weak coding propensity by chance, is rejected by the Mann-Whitney U test, which demonstrates that the distribution of Z-curve scores is sufficiently different from that of distal 3' UTRs not to occur by chance (*p* = 1.02 × 10^-23^, Mann-Whitney U test, distal 3' UTR vs novel extensions).

*Finally, you look at* D. melanogaster *SNPs to evaluate whether there is a preference for synonymous relative to non-synonymous SNPs in novel vs predicted read-through, finding that there is a weak preference for synonymous SNPs, less than found in predicted SNPs, and say this is consistent with “mild or recently-imposed selection”. This might not be correct. If you have two read-through events – one which evolved at the base of the genus* Drosophila*, and one along the lineage that separates* D. melanogaster *from* D. simulans*/*D. sechellia *– and posit that these read-through events are under identical selective pressure, then you would expect both to have identical preference for synonymous substitutions, regardless of when you evolved. There is good reason to assume that recently evolved sequences would be under any different strength of selection. If this novel hypothesis were correct, you would probably expect some, if not most, of the novel events to be polymorphic within the population, with some of these in the middle of selective sweeps. But, this would produce a different synonymous vs non-synonymous pattern (with an excess of non-synonymous SNPs perhaps)*.

*It is odd to argue that the majority of read-through events are novel and under selection – something that you'd only expect to find if readthrough events had very short evolutionary half-lives or if there were some reason to have specifically evolved functional read-through events in* D. melanogaster.

We regret that the language in our manuscript appears to have been unclear on another critical point: we did not intend to imply that the majority of extensions are both evolutionarily novel and under purifying selection, as this would yield the surprising and unlikely conclusions noted by the reviewers. Rather, we intended to state: 1) that the majority of extensions are not phylogenetically conserved as measured by PhyloCSF, and 2) that a subset of this group includes evolutionarily recent extensions that are under selection for protein coding.

The non-conserved set of extensions also includes other subsets, for example: 1) selectively neutral extensions subject to genetic drift, 2) novel extensions undergoing diversifying selection, and 3) deleterious extensions presumably not undergoing fixation and occurring only in a small subset of the population. The weak preference for synonymous SNPs among the novel extensions can be explained by the fact that this group is a heterogeneous mix of these various subsets, which themselves are subject to different magnitudes and directions of selective pressure. However, there need only be a subset of extensions undergoing purifying selection in order to make the preference deviate from the background level observed in distal 3' UTRs as we observed in our data. Granted, in order to be consistent with our observations, the effect yielded by this subset must exceed the contribution from the set of extensions undergoing diversifying selection. We have adjusted the language in our manuscript accordingly.

In addition, we have adjusted our Discussion section, to highlight our interpretation that only a subset of novel extensions are under protein-coding selection.

*The data are equally, if not more, consistent with the novel readthrough being subject to weak selection, with their origins unknown. One simple way to resolve this might be to run the Z-score program on orthologous UTRs of read-through and non-read-through genes across the genus. If these are novel to* D. melanogaster*, their scores should be significantly higher in* D. melanogaster.

We are indeed interested in more precisely determining the phylogenetic ages of the extensions we report. We believe this will be possible as more insect species are sequenced and incorporated into the insect phylogeny, and as more individuals of *Drosophila melanogaster* are sequenced and their phylogenetic relationships modeled.

At present, it is present possible to set loose bounds on the evolutionary ages of the phylogenetically conserved extensions by running PhyloCSF on phylogenetic subtrees on the sequenced *Drosophila* phylogeny, with the expectation that PhyloCSF scores for a given extension should increase as those lineages that lack the extension are pruned from the tree. However, it is not possible to perform such an analysis on novel extensions that came under selection within *melanogaster,* as these will not be conserved between species and therefore will not be measurable by cross-species tools such as PhyloCSF.

Unfortunately, a cross-species Z-curve analysis is also unlikely to answer this question, because: 1) Z-curve scores, evaluated for different species on the classifier trained specifically for *melanogaster,* would be differentially affected species-wise by species-specific nucleotide composition biases and 2) Z-curve scores for different species evaluated on different classifiers trained individually on each species will not be directly comparable, as Z-curve scores — like neural network scores or SVM scores — lack well-defined theoretical interpretations outside the score distribution of elements scored by the same classifier.

The best way to approach this would be to develop a PhyloCSF-like tool that would work on single-species population genetic data. Unfortunately, such a tool has not yet been published. While we are interested in this analysis for future work, we believe that estimating the precise evolutionary origin of each extension is beyond the purview of this specific manuscript.

Nonetheless, we do believe that, for the sake of the evolutionary model we have presented here, it is sufficient to demonstrate that a subset of the C-terminal extensions we have identified by ribosome profiling are evolutionarily novel and/or unique to *melanogaster,* even if their precise dates of origin are not defined. For the reasons outlined above (and in our manuscript), we continue to believe this interpretation to be the most plausible and consistent with our observations, and we hope the aforementioned changes to the manuscript are in the reviewers' opinions sufficient.